# AIRE-PRUNE: ASYMPTOTIC IMPULSE-RESPONSE ENERGY FOR STATE PRUNING IN STATE SPACE MODELS

**Apurba Prasad Padhy[1], Fernando Camacho[2] & Saibal Mukhopadhyay[1]**
[1] Georgia Institute of Technology, Atlanta, USA
[2] Laboratory for Physical Sciences, College Park, Maryland, USA
[1] apadhy9@gatech.edu, saibal.mukhopadhyay@ece.gatech.edu
[2] fercamacho@lps.umd.edu

## ABSTRACT

State space models (SSMs) often sacrifice capacity, search space, or stability to offset the memory and compute costs of large state dimensions. We introduce a structured post-training pruning method for SSMs — AIRE-Prune (Asymptotic Impulse- Response Energy for State PRUN(E)ing ) — that reduces each layer's state dimension by directly minimizing long-run output-energy distortion. AIRE-Prune assigns every state a closed-form *asymptotic impulse-response energy* based score, i.e., the total impulse-response energy it contributes over an infinite horizon (time), and normalizes these scores layer-wise to enable global cross-layer comparison and selection. This extends modal truncation from single systems to deep stacks and aligns pruning with asymptotic response energy rather than worst-case gain. Across diverse sequence benchmarks, AIRE-Prune reveals substantial redundancy in SISO and MIMO SSMs with average pruning of 60.8% , with average accuracy drop of 0.29% without retraining while significantly lowering compute. Code will be released: https://github.com/falcon-arrow/AIRE-Prune.

## 1 INTRODUCTION

Deep state space models (SSMs) have proven effective in modeling sequential data by optimally compressing input history into internal states (Gu et al., 2020; 2021; 2022b; Gu & Dao, 2023; Zhang et al., 2023; Parnichkun et al., 2024). Alongside these advances, a persistent challenge is to train SSMs efficiently and stably without divergence. Building on classical linear systems theory (Kailath, 1980), recent work has developed stability-guaranteeing parameterizations (Gu et al., 2022b), general architectural blueprints (Smith et al., 2023), and frequency-domain implementations that leverage transfer functions and FFTs for throughput (Gu et al., 2022a; Zhang et al., 2023; Parnichkun et al., 2024).

A central driver of computation and memory in SSMs is the *state dimension $n$*. Since early proposals, a multiple single-input single-output (multi-SISO) construction has been widely adopted for scalable training: many small SISO subsystems are learned in parallel and then mixed through channel projections (Gu et al., 2022b; Gu & Dao, 2023; Zhang et al., 2023; Parnichkun et al., 2024). Within this setting, diagonal (or diagonalizable) systems have been shown to match the accuracy of more general non-diagonal systems while remaining efficient [Gupta et al., 2022]. In parallel, multi-input multi-output (MIMO) SSMs (e.g., S5-style stacks) explicitly exploit multi-channel structure, often attaining strong performance with a smaller $n$ than equivalently sized multi-SISO blocks; this advantage is especially evident on long-horizon tasks such as Path-X (Smith et al., 2023; Parnichkun et al., 2024) . Despite these developments, both multi-SISO and MIMO families typically lack *post-training mechanisms* to optimize $n$, leading to over-parameterization and unnecessary inference cost.

Several orthogonal directions attempt to reduce complexity including unstructured weight pruning or structured state pruning strategies. This work focuses on structured state pruning as it translates to real computation reduction with reduced dimensionality. Parameterizing transfer functions for SISO systems can enable state-free inference, but tends to restrict the search space or guarantee stability only at initialization (Parnichkun et al., 2024). Frequency-domain kernels accelerate training and inference but do not directly resolve redundancy in the learned state space (Gu et al., 2022a; Gupta

et al., 2022; Zhang et al., 2023). This motivates *post-training model order reduction* (MOR) in deep SSMs: identify and remove states that minimally affect the task output while preserving accuracy.

Layer-adaptive pruning provides a practical MOR route. Recent work (LAST) (Gwak et al., 2025) proposed to score each learned subsystem by a worst-case, frequency-domain gain (an $H_\infty$ view), then normalize scores layer-wise to enable global, cross-layer selection and pruning with bounded output distortion. Empirically, this revealed that trained SSMs are often highly compressible, achieving sizable state reductions with minimal accuracy loss on long-range sequence benchmarks (e.g., LRA and Speech Commands). While powerful, worst-case measures can be conservative for typical workloads, as they emphasize peak amplification that tasks may rarely excite.

**This work.** We introduce **AIRE-Prune** (Asymptotic Impulse- Response Energy for State PRUN(E)ing), a structured, post-training, layer-adaptive method for pruning states in deep state space model. The contribution of this work are as follows:

- **Energy-based ranking:** ranks each state (mode) by its *total output energy* over infinite time when excited under a *unit impulse input*.
- **Closed-form per-mode energy score:** assigns every learned mode a *closed-form* total output energy score, enabling fast, principled importance ordering.
- **Cross-layer normalization ⇒ global pruning score:** normalizes per-layer energy scores to a common scale, enabling *global, cross-layer* comparison and selection of the *least-energetic* (insignificant) states.

This extends classical modal truncation from single systems to deep stacks and aligns pruning with *asymptotic impulse response energy*—a typical-case criterion—rather than a worst-case gain approach. AIRE-Prune is architecture-agnostic (covering multi-SISO and MIMO SSMs).

**Results.** On Long Range Arena with S5-style MIMO SSMs, AIRE-Prune *prunes on average* **60.8%** of states with only **0.29%** average accuracy degradation *without retraining*. These findings indicate that state spaces in trained SSMs contain substantial removable redundancy and that aligning pruning with total output energy yields strong compression at negligible cost to task performance, surpassing worst-case frequency gain-based approaches.

## 2 RELATED WORK

### 2.1 MODEL ORDER REDUCTION (MOR)

MOR approximates high-dimensional linear systems by lower-order models with controlled error, with applications in VLSI (Antoulas & Sorensen, 2001), power systems (Li & White, 1999), and PDE discretizations (Jones & Kerrigan, 2010; Curtain & Zwart, 2012). *Modal truncation* removes states from a diagonal realization using frequency-domain criteria that limit $H_\infty$ distortion (Green & Limebeer, 2012). *Balanced truncation* constructs a realization in which states are simultaneously controllable and observable, then discards the least energetic ones, offering strong error guarantees and many variants (Jones & Kerrigan, 2010; Curtain & Zwart, 2012). However, the similarity transforms central to balanced truncation eliminate the diagonal structure exploited by modern SSMs for efficiency and stability. **Our stance:** we preserve diagonal parameterization and *per-state* granularity (as in modal truncation) and extend the setting from a single linear system to *deep stacks with nonlinearities* encountered in contemporary sequence models.

### 2.2 LAYER-ADAPTIVE PRUNING IN DEEP NETWORKS

Allowing different layers to prune by different amounts (layer-adaptive pruning) (Morcos et al., 2019; Han et al., 2015; Mocanu et al., 2018; Lee et al., 2021; Xu et al., 2023) generally outperforms uniform ratios (Zhu & Gupta, 2017; Gale et al., 2019). Approaches include layerwise magnitude thresholds/targets and *global* criteria that compare scores across layers under a single budget. For example, provides a Frobenius-norm bound on worst-case $\ell_2$ distortion when pruning one layer (others frozen), and extends this to joint optimization over layerwise ratios. Because SSMs are governed

by transfer functions and dynamical couplings—not only static weights—a *non-magnitude* importance criterion is essential. **Our stance:** we adopt the layer-adaptive paradigm but base importance on *total output energy*, yielding closed-form, mode-separable scores that respect controllability, observability, and damping.

## 2.3 STATE PRUNING FOR DEEP SSMS: LAST VS. THIS WORK

LAST (Gwak et al., 2025) ranks states via a *worst-case* frequency-domain measure: per-state $H_\infty$ scores are normalized to produce a *global* cross-layer ranking under a model-level budget, giving a *peak-gain* bound on output distortion and showing trained SSMs are highly compressible. **This work (AIRE-Prune):** we retain per-state, layer-adaptive pruning but use *infinite-horizon output energy* (total impulse-response energy) as a *typical-case* criterion that emphasizes long-run expenditure through controllability/observability/damping. This energy view yields closed-form, layer-normalized, globally comparable scores and, in practice, enables aggressive compression with negligible degradation in accuracy. Refer appendix B.6, for detailed mathematical difference.

# 3 BACKGROUND

## 3.1 STABILITY OF STATE SPACE MODELS

A DT LTI SSM is asymptotically stable iff its poles lie strictly inside the unit circle, i.e., $\rho(A) < 1$. Directly enforcing this during training is nontrivial: constraining parameters to a fixed stable set (e.g., disk/polygon) guarantees stability but can reduce expressivity, while "center-of-stability" initialization helps early training yet offers no guarantee thereafter. Diagonal (or diagonalizable) SSMs mitigate this by *parameterizing poles directly*: we use a diagonal DT realization with conjugate pairs for real I/O. When derived from CT diagonal poles with $\Re(\lambda_i^{\mathrm{ct}}) < 0$, zero-order hold at step $\Delta_i$ gives $\lambda_i = \exp(\lambda_i^{\mathrm{ct}} \Delta_i)$, hence $|\lambda_i| < 1$. This diagonal parameterization preserves stability by construction and enables classical analyses (Lyapunov, modal formulas) within deep SSM layers.

## 3.2 DIAGONAL STATE SPACE MODELS (S5-STYLE)

**High-level architecture.** An S5 layer processes a length-$T$ sequence $u_{0:T-1} \in \mathbb{R}^h$ through a diagonal, linear time-invariant (LTI) core followed by a pointwise nonlinearity and channel mixing. As in prior diagonal SSMs, the stack comprises: an *encoder* that lifts inputs to $h$ channels, a sequence of *LTI+nonlinearity* blocks, and a *decoder* tailored to the downstream task. Each block can be implemented either as a *multi-SISO* assembly (one diagonal SISO system per channel) or as a single *MIMO* system acting jointly on the $h$ channels. We adopt a unified MIMO description whose *effective* state dimension per layer is

$$n = \begin{cases} n_s\, h & \text{for multi-SISO ($n_s$ states per channel)} \\ n_m & \text{for MIMO ($n_m$ shared states)} \end{cases} \tag{1}$$

so that multi-SISO is a structured special case of MIMO with block-diagonal couplings.

**Parameterization.** We model each diagonal SSM layer (state size $n$; channels $h$) in continuous time (CT) by $\boldsymbol{x}(t) = \boldsymbol{\Lambda}\boldsymbol{x}(t) + \boldsymbol{B}\boldsymbol{u}(t)$ and $\boldsymbol{y}(t) = \boldsymbol{C}\boldsymbol{x}(t) + \boldsymbol{D}\boldsymbol{u}(t)$, where $\boldsymbol{\Lambda} = \mathrm{diag}(\lambda_1, \ldots, \lambda_n) \in \mathbb{C}^{n \times n}$ is diagonal, $\boldsymbol{B} \in \mathbb{C}^{n \times h}$, $\boldsymbol{C} \in \mathbb{C}^{h \times n}$, and $\boldsymbol{D} \in \mathbb{R}^{h \times h}$. To preserve real inputs/outputs, complex parameters appear in conjugate pairs (Gu et al., 2022b). Following diagonal SSM practice, each mode $i$ carries its own step size $\Delta_i > 0$ (collected into $\Delta \in \mathbb{R}^n$), so zero-order hold (ZOH) discretization yields a diagonal DT system

$$\boldsymbol{x}_{k+1} = \boldsymbol{\Lambda}\boldsymbol{x}_k + \boldsymbol{B}\boldsymbol{u}_k, \qquad \boldsymbol{y}_k = \boldsymbol{C}\boldsymbol{x}_k + \boldsymbol{D}\boldsymbol{u}_k, \tag{2}$$

$$where, \quad \boldsymbol{\Lambda}_d = \mathrm{diag}\!\left(e^{\lambda_i \Delta_i}\right), \qquad \boldsymbol{B}_d = \boldsymbol{\Lambda}^{-1}\!\left(\boldsymbol{\Lambda}_d - I\right)\boldsymbol{B}, \tag{3}$$

where the $\boldsymbol{B}_d$ expression holds elementwise and extends continuously at $\lambda_i = 0$. Stability is enforced at the CT level (Hurwitz), i.e., $\Re(\lambda_i) < 0$, which implies $|e^{\lambda_i \Delta_i}| < 1$ and thus DT contractivity. A pointwise nonlinearity wraps the linear core to form the layer output, $f_\sigma(\boldsymbol{u}_k; \Sigma) =$

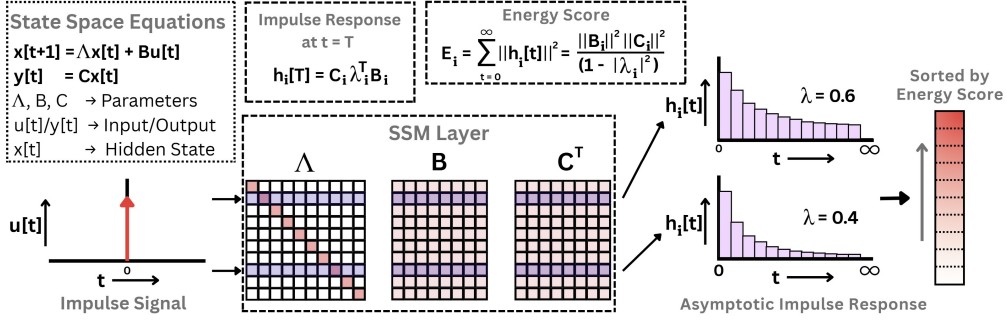

Figure 1: AIRE-Prune: Asymptotic Impulse Response Local Energy Score

$\sigma\big(\boldsymbol{C}\boldsymbol{x}_k + \boldsymbol{D}\boldsymbol{u}_k\big)$, optionally combined with residual and normalization layers. This diagonal parameterization admits closed-form per-mode discretization, simple stability control, and efficient kernel/scan implementations, while remaining applicable to both multi-SISO ($n = n_s h$) and MIMO ($n = n_m$) instantiations under a unified notation.

## 4 AIRE-PRUNE: ASYMPTOTIC IMPULSE-RESPONSE ENERGY FOR STATE PRUNING IN STATE SPACE MODELS

**State space model.** We begin with a discrete-time (DT) diagonal state space layer

$$\boldsymbol{x}_{k+1} = \boldsymbol{\Lambda}\boldsymbol{x}_k + \boldsymbol{B}\boldsymbol{u}_k, \qquad \boldsymbol{y}_k = \boldsymbol{C}\boldsymbol{x}_k, \tag{4}$$

where $\boldsymbol{\Lambda} = \mathrm{diag}(\lambda_1, \dots, \lambda_n)$ with $|\lambda_i| < 1$ (all poles strictly inside the unit circle). This spectral condition is the DT counterpart of asymptotic stability in control: it guarantees that perturbations decay and that linear system quantities defined as *infinite sums over time* are finite. Diagonal (or diagonalizable) parameterizations make stability explicit through pole variables $\{\lambda_i\}$ and enable mode-wise analysis.

**Why energy as importance metric?** In linear time-invariant (LTI) systems, "importance" has a canonical meaning: *how much output energy a direction (state) can transmit from inputs to outputs*. For stable discrete-time systems this energy is literally an *area under a curve*, over time (sum of squared impulse response) or over frequency (integral of squared transfer magnitude). This section develops that basic connection and, from it, derives a simple state-importance score and a practical pruning rule. An impulse (or white noise) excites all frequencies equally, so the measured energy reflects *aggregate* amplification (realizable to real input variations), not a single resonant tone.

For total (steady-state) output energy corresponding to a state, we consider a stable discrete-time SSM layer $\Sigma : (\boldsymbol{\Lambda}, \boldsymbol{B}, \boldsymbol{C})$ with zero initial state and an impulse input $u_k = \delta_{k0}$. The impulse response is

$$\boldsymbol{H}_k = \boldsymbol{C}\boldsymbol{\Lambda}^k \boldsymbol{B},$$

and the output is $\boldsymbol{y}_k = \boldsymbol{H}_k$. The (squared) *steady-state output energy* is the discrete-time $\mathcal{H}_2$ energy

$$\|\Sigma\|_{\mathrm{energy}}^2 \;=\; \underbrace{\frac{1}{2\pi} \int_0^{2\pi} \|\boldsymbol{G}(e^{j\omega})\|_F^2 \, d\omega}_{\text{frequency domain}} \;=\; \underbrace{\sum_{k=0}^{\infty} \|\boldsymbol{H}_k\|_F^2}_{\text{time domain}} \;=\; \sum_{k=0}^{\infty} \|\boldsymbol{C}\boldsymbol{\Lambda}^k \boldsymbol{B}\|_F^2, \tag{5}$$

where the transfer function is $\boldsymbol{G}(z) = \boldsymbol{C}(z\boldsymbol{I} - \boldsymbol{\Lambda})^{-1}\boldsymbol{B}$, and Parseval's identity yields the equality of time- and frequency-domain energies. This admits two equivalent interpretations that connect directly to control and signal processing: (i) *impulse energy*: the total squared output produced by a unit impulse, accumulated from $k = 0$ to $\infty$; and (ii) *long-run output power under unit white noise*: the steady-state variance of $\boldsymbol{y}_k$ when $u_k \sim \mathcal{N}(0, \boldsymbol{I})$ (by Parseval's identity). Both views rely on $|\lambda_i| < 1$ to ensure that the series in equation 5 converges, which is the case for the diagonal SSMs considered here.

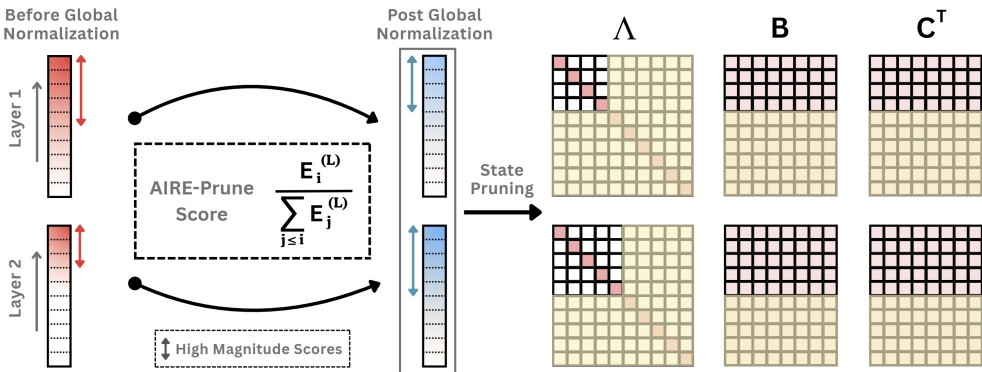

Figure 2: AIRE-Prune: Prefix-normalized global scoring across all the layers. (Yellow shade defines pruned states as they correspond to low magnitude score)

**Per-mode energy via a geometric progression (finite horizon).** For a single diagonal mode $\Sigma_i$ : $(\lambda_i, \boldsymbol{B}_{i,:}, \boldsymbol{C}_{:,i})$ with $|\lambda_i| < 1$, the rank-1 impulse slice is $\boldsymbol{H}_t^{(i)} = \boldsymbol{C}_{:,i} \lambda_i^t \boldsymbol{B}_{i,:}$. Using $\|\boldsymbol{u}\boldsymbol{v}^\top\|_F^2 = \|\boldsymbol{u}\|_2^2 \|\boldsymbol{v}\|_2^2$ for rank-1 outer products,

$$\left\|\boldsymbol{H}_t^{(i)}\right\|_F^2 = |\lambda_i|^{2t} \|\boldsymbol{C}_{:,i}\|_2^2 \|\boldsymbol{B}_{i,:}\|_2^2. \tag{6}$$

Therefore, the (truncated) output energy contributed by mode $i$ over horizon $T$ is the geometric sum

$$E_i(T) = \sum_{t=0}^{T-1} \left\|\boldsymbol{H}_t^{(i)}\right\|_F^2 = \|\boldsymbol{C}_{:,i}\|_2^2 \|\boldsymbol{B}_{i,:}\|_2^2 \sum_{t=0}^{T-1} |\lambda_i|^{2t} = \|\boldsymbol{C}_{:,i}\|_2^2 \|\boldsymbol{B}_{i,:}\|_2^2 \frac{1 - |\lambda_i|^{2T}}{1 - |\lambda_i|^2} . \tag{7}$$

Stability of the state space model implies $|\lambda_i|^{2T} \to 0$ as $T \to \infty$. Taking the limit in equation 7 gives the *per-mode steady-state output energy which is equivalent to the impulse response energy.*

$$E_i = \lim_{T \to \infty} E_i(T) = \frac{\|\boldsymbol{C}_{:,i}\|_2^2 \|\boldsymbol{B}_{i,:}\|_2^2}{1 - |\lambda_i|^2} , \tag{8}$$

and the total layer energy can be approximated as an additive sum across modes (Appendix A.1.2),

$$\|\Sigma\|_{\text{energy}}^2 \approx \sum_{i=1}^n E_i.$$

Pruning a set $P$ of states removes their modal responses. The resulting change in layer energy is approximately the sum of their individual contributions:

$$\|\Sigma\|_{\text{energy}}^2 - \|\Sigma_{-P}\|_{\text{energy}}^2 \approx \sum_{i \in P} E_i. \tag{9}$$

Thus, to minimize steady-state distortion at the layer level, one should remove the *smallest-energy* modes. We therefore define the **energy (AIRE-Prune local) score** for state $x_i$ as

$$\text{EnergyScore}_{\text{local}}(x_i) = E_i = \frac{\|\boldsymbol{C}_{:,i}\|_2^2 \|\boldsymbol{B}_{i,:}\|_2^2}{1 - |\lambda_i|^2} . \tag{10}$$

*Special cases.* If $\boldsymbol{B}$ is fixed and row-normalized (common in practice), then $\|\boldsymbol{B}_{i,:}\|_2 = 1$ and $E_i = \|\boldsymbol{C}_{:,i}\|_2^2/(1 - |\lambda_i|^2)$. For bidirectional layers, $\|\boldsymbol{C}_{:,i}\|_2^2$ is replaced by the average of the forward and backward contributions.

| Method | State importance |
|---|---|
| Random | $-$ |
| Uniform magnitude | $\lvert\bar\lambda_i\rvert\,\lVert\bar{\mathbf{B}}_i\rVert\,\lVert\mathbf{C}_i\rVert$ |
| Global magnitude | $\lvert\bar\lambda_i\rvert\,\lVert\bar{\mathbf{B}}_i\rVert\,\lVert\mathbf{C}_i\rVert$ |
| LAMP | $\dfrac{\lvert\bar\lambda_i\rvert^2\,\lVert\bar{\mathbf{B}}_i\rVert^2\,\lVert\mathbf{C}_i\rVert^2}{\sum_{j\le i}\lvert\bar\lambda_j\rvert^2\,\lVert\bar{\mathbf{B}}_j\rVert^2\,\lVert\mathbf{C}_j\rVert^2}$ |
| Uniform $\mathcal{H}_\infty$ | $\dfrac{\lVert\mathbf{C}_i\rVert^2\,\lVert\bar{\mathbf{B}}_i\rVert^2}{(1-\lvert\bar\lambda_i\rvert)^2}$ |
| Global $\mathcal{H}_\infty$ | $\dfrac{\lVert\mathbf{C}_i\rVert^2\,\lVert\bar{\mathbf{B}}_i\rVert^2}{(1-\lvert\bar\lambda_i\rvert)^2}$ |
| LAST | $\dfrac{\frac{\lVert\mathbf{C}_i\rVert^2\,\lVert\bar{\mathbf{B}}_i\rVert^2}{(1-\lvert\bar\lambda_i\rvert)^2}}{\sum_{j\le i}\frac{\lVert\mathbf{C}_j\rVert^2\,\lVert\bar{\mathbf{B}}_j\rVert^2}{(1-\lvert\bar\lambda_j\rvert)^2}}$ |
| **AIRE-Prune (ours)** | $\dfrac{\frac{\lVert\mathbf{C}_i\rVert^2\,\lVert\bar{\mathbf{B}}_i\rVert^2}{1-\lvert\bar\lambda_i\rvert^2}}{\sum_{j\le i}\frac{\lVert\mathbf{C}_j\rVert^2\,\lVert\bar{\mathbf{B}}_j\rVert^2}{1-\lvert\bar\lambda_j\rvert^2}}$ |

(a) State-importance definitions

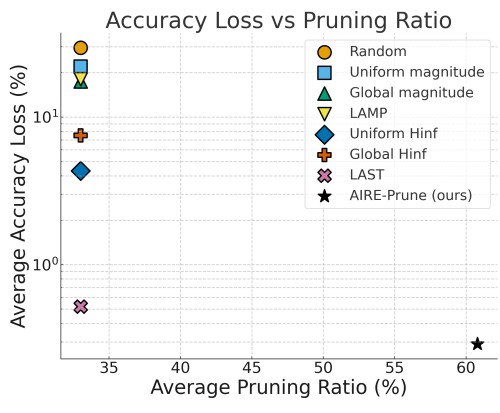

(b) Accuracy loss vs. pruning ratio.

Figure 3: Comparison across baselines

**Intuition: Why this score is a good importance measure.** The score in Eq. equation 10 is (a) *mode-separable* (no large matrix solves), (b) *scale-aware* through the $\boldsymbol{B}/\boldsymbol{C}$ couplings, and (c) *dynamics-aware* through the pole damping. A state with larger $\lVert\tilde{\boldsymbol{B}}_{i,:}\rVert_2$ (more controllable, i.e., easier to excite), larger $\lVert\tilde{\boldsymbol{C}}_{:,i}\rVert_2$ (more observable, i.e., easier to measure), or larger $\lvert\lambda_i\rvert$ (longer memory) has larger $E_i$ and therefore induces a larger accuracy or variance drop if removed. This aligns pruning with steady-state distortion and empirically enables aggressive compression with negligible accuracy loss.

**From local to global: layer normalization.** Per-mode asymptotic energies $E_i^{(\ell)}$ can differ in scale across layers (e.g., due to encoder/decoder gains), therefore we need to normalize individual layers in such a way to have a common global threshold for state pruning. The per-layer scales can differ substantially, so we first sort the states of each layer by their Per-mode Asymptotic Energy in descending order. Let $E_{(i)}^{(\ell)}$ denote the $i$-th largest per-mode asymptotic energy in layer $\ell$, and define the *prefix sum*

$$S_{(i)}^{(\ell)} \;=\; \sum_{j\le i} E_{(j)}^{(\ell)}.$$

We then use the **prefix-normalized score (AIRE-Prune)**

$$\mathrm{AIREPrune}\big(x_{(i)}^{(\ell)}\big) \;=\; \frac{E_{(i)}^{(\ell)}}{S_{(i)}^{(\ell)} + \varepsilon}, \tag{11}$$

with a small $\varepsilon > 0$ for numerical stability. This "hazard-rate" ratio is *monotonically non-increasing* in $i$, enabling an *elbow-style, layer-adaptive* rule: given a single global threshold $\tau$, we *keep* the longest prefix in each layer for which $\mathrm{Score}_{\text{prefix}} \ge \tau$ and prune the remaining contiguous tail.

**Summary.** Stability ensures that infinite-horizon energy is well-posed; energy decomposes additively by modes in diagonal SSMs; the closed-form per-mode energy equation 10 gives an interpretable, dynamics- and scale-aware *local* importance; and simple *layer normalization* equation 11 turns these into globally comparable scores for cross-layer sensitive pruning. Refer Appendix for mathematical analysis of the above.

---

**Algorithm 1:** AIRE-Prune: Asymptotic Impulse-Response Energy for State Pruning

---

**Input:** SSM Layers $\{\ell = 1 \dots L\}$ with parameters $(\mathbf{\Lambda}^{(\ell)}, \boldsymbol{B}^{(\ell)}, \boldsymbol{C}^{(\ell)})$; prune ratio $p$.
**Output:** Kept-index sets $\{\mathcal{K}^{(\ell)}\}$ and pruned sets $\{\mathcal{P}^{(\ell)}\}$.

1 **Step 1: Per-mode asymptotic energy $E_i^{(\ell)}$.**
2 **for** $\ell \leftarrow 1$ **to** $L$ **do**
3      Extract modal triplets $\{(\lambda_i^{(\ell)}, \boldsymbol{B}_{i,:}^{(\ell)}, \boldsymbol{C}_{:,i}^{(\ell)})\}_{i=1}^{n_\ell}$ with $|\lambda_i^{(\ell)}| < 1$.
4      $E_i^{(\ell)} \leftarrow \dfrac{\|\boldsymbol{C}_{:,i}^{(\ell)}\|_2^2 \, \|\boldsymbol{B}_{i,:}^{(\ell)}\|_2^2}{1 - |\lambda_i^{(\ell)}|^2}$

5 **Step 2: Per-layer sorting and prefix sums.**
6 **for** $\ell \leftarrow 1$ **to** $L$ **do**
7      Sort modes by $E_i^{(\ell)}$ in descending order to get $E_{(1)}^{(\ell)} \geq \cdots \geq E_{(n_\ell)}^{(\ell)}$ and their indices.
8      Compute prefix sums: $S_{(i)}^{(\ell)} \leftarrow \sum_{j \leq i} E_{(j)}^{(\ell)}$ for $i = 1 \dots n_\ell$.
9      Compute AIRE-Prune scores: $s_{(i)}^{(\ell)} \leftarrow \dfrac{E_{(i)}^{(\ell)}}{S_{(i)}^{(\ell)} + \varepsilon}$ where $\varepsilon \to 0$

10 **Step 3: Global selection**
11 **if** *prune ratio $p$ given* **then**
12      $B \leftarrow \sum_{\ell=1}^{L} n_\ell \cdot (1 - p)$
13 Sort AIRE-Prune in decending order to get global list
     $\mathcal{L} \leftarrow \{(s_{(i)}^{(\ell)}, \ell, i) \mid 1 \leq \ell \leq L, \ 1 \leq i \leq n_\ell\}$.
14 Find the value $\tau$ equal to the $B$-th largest score in $\mathcal{L}$ Define the kept count per layer by
     $k_\ell \leftarrow \max\{i : s_{(i)}^{(\ell)} \geq \tau\}$

15 **Step 4: Materialize keep/prune indices.**
16 **for** $\ell \leftarrow 1$ **to** $L$ **do**
17      $\mathcal{K}^{(\ell)} \leftarrow$ original indices of the top $k_\ell$ sorted modes;    $\mathcal{P}^{(\ell)} \leftarrow$ Pruned indices.
18 **return** $\{\mathcal{K}^{(\ell)}\}, \{\mathcal{P}^{(\ell)}\}$

---

## 5 EXPERIMENTS

### 5.1 SETUP

**Models and tasks.** We evaluate pruning on the **S5** (MIMO) SSM (Smith et al., 2023) across the six Long Range Arena (LRA) tasks (Tay et al., 2021) and the **Speech Commands** 35-way keyword recognition benchmark (Warden, 2018) (sequence length 16,000). All runs use a single **NVIDIA H100** (40GB/80GB) GPU with the released S5 training and inference configuration. Unless noted otherwise, we report one-shot pruning (no retraining) and evaluate accuracy by freezing all parameters.

**Baselines.** We compare against six pruning baselines which includes random, uniform magnitude, global magnitude, LAMP (Lee et al., 2021), Uniform $H_\infty$ (Gwak et al., 2025), Global $H_\infty$ (Gwak et al., 2025), LAST (Gwak et al., 2025). Figure 3a shows the state importance metric (scoring method) for the baselines.

**Additional architectures.** Beyond S5, we also evaluate AIRE-PRUNE on **S4D** and **Mamba (S6)** backbones to test generalization across SSM families. Task-wise pruning/accuracy numbers for these models on LRA are summarized in Table 2.

**Pruning ratios.** For methods that allocate layerwise budgets (Global $H_\infty$, AIRE-Prune), we report *average* pruning ratios across layers. We sweep pruning ratios from $\{0\% \text{ to } 100\%\}$ with step size

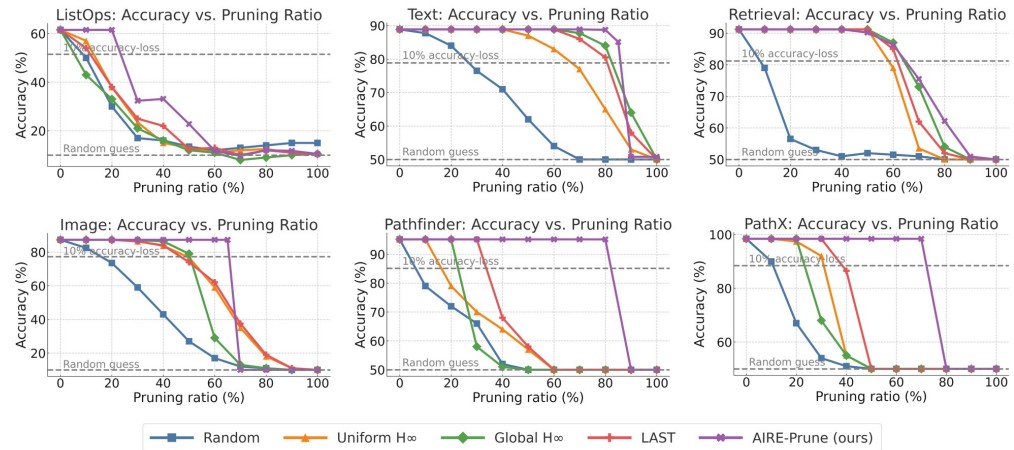

Figure 4: Trade-off curves between pruning ratio and accuracy for pruned S5 models across tasks in the LRA benchmark. Baselines LAST, Uniform $H_\infty$, Global $H_\infty$ are refered from (Gwak et al., 2025)

Table 1: **Task-wise accuracy on LRA and Speech Commands with S5.** Pruning ratio (Prun.) and post-pruning accuracy (Acc.). Numbers for Full/Uniform $H_\infty$/Global $H_\infty$/LAST (Gwak et al., 2025) are compared with AIRE-Prune (ours).

| Model | ListOps (2,048) | | Text (4,096) | | Retrieval (4,000) | | Image (1,024) | | Pathfinder (1,024) | | Path-X (16,384) | | Speech (16,000) | |
|---|---|---|---|---|---|---|---|---|---|---|---|---|---|---|
| | Prun. | Acc. | Prun. | Acc. | Prun. | Acc. | Prun. | Acc. | Prun. | Acc. | Prun. | Acc. | Prun. | Acc. |
| **S5 Full** | 0% | 61.48 | 0% | 88.88 | 0% | 91.20 | 0% | 87.30 | 0% | 95.15 | 0% | 98.41 | 0% | 96.43 |
| Uniform $H_\infty$ | 0% | 61.48 | 60% | 82.49 | 50% | 90.29 | 30% | 86.45 | 30% | 71.38 | 30% | 90.90 | 20% | 96.20 |
| Global $H_\infty$ | 0% | 61.48 | 60% | 88.56 | **50%** | **90.93** | 30% | 87.04 | 30% | 57.20 | 30% | 69.21 | 20% | 96.21 |
| LAST | 0% | 61.48 | 60% | 88.52 | 50% | 90.42 | 30% | 86.34 | 30% | 94.45 | 30% | 97.95 | 20% | 96.31 |
| **AIRE-Prune (ours)** | **20%** | **61.48** | **80%** | **88.24** | 50% | 90.11 | **65%** | **87.30** | **80%** | **95.15** | **70%** | **98.41** | **45%** | **96.40** |

of 10% for Uniform $H_\infty$ (Gwak et al., 2025), Global $H_\infty$ (Gwak et al., 2025), LAST (Gwak et al., 2025) and AIRE-Prune (our work) as in Figure 4. A ratio of 100% would leave a single complex-conjugate pair per layer.

## 5.2 ANALYSIS

**Long Range Arena (S5).** We evaluate AIRE-Prune on the LRA suite (Tay et al., 2021), which probes long-range dependencies with sequence lengths from 1,024 to 16,384. In all runs we apply a single global threshold to the prefix-normalized AIRE scores and prune without any retraining. Table 1 reports the post-pruning accuracies.

**Speech Commands (S5).** On Speech Commands, AIRE-PRUNE achieves **45%** state pruning while essentially preserving accuracy (96.40% vs. 96.43% for the full S5 model; Table 1). This mirrors the high-compressibility regime observed on several LRA tasks and indicates that AIRE's energy-based ranking transfers cleanly from synthetic long-range benchmarks to real audio classification.

**Aggregate outcome.** Averaged across tasks, AIRE-Prune removes **60.8%** of states with only **0.29 pp** accuracy drop (Figure 3b, last row). In the common "≤1 pp loss" regime, compressibility is task-dependent: *Text*, *Pathfinder*, and *Path-X* sustain **80%**, **80%**, and **70%** pruning, respectively, while matching or nearly matching full accuracy (Text: 88.24 vs. 88.88; Pathfinder: 95.15 vs. 95.15; Path-X: 98.41 vs. 98.41; Table 1). *Retrieval* and *Image* tolerate **50-65%** pruning at ≤1 pp loss (Re-

Table 2: **Task-wise accuracy on LRA with S4D and Mamba.** Pruning ratio (Prun.) and post-pruning accuracy (Acc.). Numbers for Full/Uniform $H_\infty$/Global $H_\infty$/LAST (Gwak et al., 2025) (S4D) are compared with AIRE-Prune (ours).

| Model | ListOps (2,048) | | Text (4,096) | | Retrieval (4,000) | | Image (1,024) | |
| | Prun. | Acc. | Prun. | Acc. | Prun. | Acc. | Prun. | Acc. |
|---|---|---|---|---|---|---|---|---|
| **S4D Full** | 0% | 56.42 | 0% | 86.40 | 0% | 90.46 | 0% | 77.02 |
| Uniform $H_\infty$ | 10% | 55.82 | 80% | 86.02 | 60% | 89.87 | 0% | 77.02 |
| Global $H_\infty$ | 10% | 49.95 | 80% | 86.20 | 60% | 89.84 | 0% | 77.02 |
| LAST | 10% | 56.27 | 80% | 85.95 | 60% | 89.46 | 0% | 77.02 |
| **AIRE-Prune (ours)** | **35%** | **56.07** | **90%** | 85.83 | -% | - | **40%** | **76.32** |
| **Mamba Full** | 0% | 38.02 | 0% | 82.98 | 0% | 72.14 | 0% | 69.82 |
| **LAST** | -% | - | 20% | 81.32 | 45% | 71.37 | 45% | 69.13 |
| **AIRE-Prune (ours)** | -% | - | **45%** | **81.19** | **60%** | **71.45** | **80%** | **68.97** |

Table 3: **Performance (S5)**: pruning ratios and resulting inference speedups and parameter-memory reductions for S5

| Metric ↓ / Dataset → | ListOps | Text | Retrieval | Image | Pathfinder | Path-X |
|---|---|---|---|---|---|---|
| **Pruning ratio** | 20% | 80% | 50% | 65% | 80% | 70% |
| **Inference speedup** ($\times$) | 1.23$\times$ | 2.49$\times$ | 1.82$\times$ | 1.93$\times$ | 2.82$\times$ | 2.86$\times$ |
| **Model reduction (%)** | 19.3% | 58.7% | 38.3% | 45.3% | 64.7% | 64.2% |

trieval: 90.11 vs. 90.93; Image: 87.30 vs. 87.30). *ListOps* is sensitive, admitting only **20%** pruning before degradation; at this operating point, accuracy is preserved (61.48 vs. 61.48).

**Comparison to baselines.** At high compression (right half of each panel in Fig. 4), AIRE-PRUNE dominates *Uniform/Global* $\mathcal{H}_\infty$ magnitude rules and outperforms LAST under the $\leq$1 pp loss criterion. This is mirrored in the *average accuracy loss* of Figure 3b: AIRE-PRUNE (0.29 pp) improves over *Uniform* $\mathcal{H}_\infty$ (4.32 pp), *Global* $\mathcal{H}_\infty$ (7.51 pp), and LAST (0.52 pp), while achieving a much higher *average* pruning ratio (60.8% vs. 33% for those methods; $\approx 1.8\times$ higher).

**Generalization to S4D and Mamba.** Table 2 shows that AIRE-PRUNE also yields competitive pruning/accuracy trade-offs on **S4D** and **Mamba**. On S4D, AIRE attains up to **90%** pruning on *Text* and **40%** on *Image* while closely tracking full-model accuracy. On Mamba, AIRE safely prunes **45%** of states on *Text*, **60%** on *Retrieval*, and **80%** on *Image* with negligible degradation. These results suggest that the asymptotic energy ranking is not tied to a specific SSM parameterization and can serve as a drop-in pruning rule across modern state-space architectures.

**Superiority over state of the art.** Against the strongest baselines (Uniform/Global $\mathcal{H}_\infty$ and LAST), AIRE-PRUNE delivers strictly larger *safe* no-retrain budgets (defined at $\leq$ 1 pp accuracy drop) on five of six LRA tasks while matching or exceeding accuracy. From Table 1, AIRE's safe pruning exceeds the best baseline by **+20 pp** on *ListOps*, **+20 pp** on *Text*, **+35 pp** on *Image*, **+50 pp** on *Pathfinder*, and **+40 pp** on *Path-X*; on *Retrieval*, the listed AIRE point at 50% misses the $\leq$ 1 pp rule by only 0.09 pp, yielding no advantage at that specific sample yet remaining competitive with the best baseline. We attribute this margin to AIRE's time-accumulated energy lens and prefix normalization, which together create strong head-tail separation and enable a single global threshold to keep essential modes intact and collapse low-energy layers, a behavior not observed in magnitude- or peak-gain-based scoring.

**Inference efficiency and model size.** We next quantify the system-level impact of pruning-by-removal. For each LRA task, we transfer the kept states into a dimension-reduced S5 model and measure inference throughput and parameter reduction on an NVIDIA H100 GPU (Table 3). Across tasks, AIRE's pruning schedules yield **1.2$\times$–2.9$\times$** speedups and **19%–65%** reductions in parameter count, with larger gains on high-pruning regimes such as *Pathfinder* and *Path-X*. This confirms that

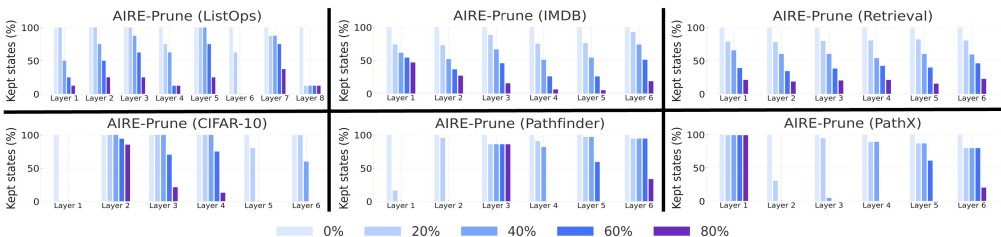

Figure 5: Layer-wise pruning ratio as we increase the global pruning threshold for S5 models across tasks in the LRA benchmark.

aggressive state pruning translates into meaningful computational and memory savings, not just abstract sparsity. Similar analysis for Mamba has been mentioned in Appendix C.3

**Accuracy drop vs. pruning ratio.** Figure 4 exhibits a pronounced *elbow/step* for *AIRE-Prune*: accuracy remains near the full model up to a high pruning threshold, then drops sharply; in contrast, baselines degrade *smoothly* as pruning increases. We interpret the step as evidence that AIRE's ranking separates *important* from *unimportant* states with a wide margin, while smooth curves indicate that pruning mixes both groups throughout the sweep. *Takeaway.* The step-like profile is robust only when using AIRE energies *and* prefix normalization, which together keep the high-margin *head* intact and prune contiguous *tails*. Baselines show smooth decay because their scores offer weaker head-tail separation, so valuable states are progressively removed throughout the sweep.

**Layer-wise pruning profile.** Figure 5 shows kept-state percentages per layer as the global ratio increases, revealing task-specific structure. On *Path-X* and *Pathfinder*, AIRE progressively thins later layers and, at high ratios, can *remove entire layers* whose states fall below the global threshold, turning fine-grained sparsity into block-level deletions. This indicates that our method can help achieve low-latency SSMs, as we are able to prune layers without losing accuracy. For *Text* and *Retrieval*, earlier layers retain larger fractions while mid/late layers contribute most of the budget, consistent with encoder-style front-end processing. *Image* exhibits more even shrinkage until its elbow, after which a few layers collapse. *ListOps* shows non-trivial contribution from all layers even at small ratios, explaining its early elbow and our conservative cap around 20%. This heterogeneous, layer-selective behavior contrasts with reported layer pruning profile of LAST (Gwak et al., 2025), which tends to have more uniform pruning among layers at similar global budgets.

## 6 CONCLUSION

We introduced AIRE-PRUNE, a training-free pruning criterion for diagonal (or diagonalized) state-space models that ranks modes by *asymptotic output energy*, combined with *prefix normalization* that puts scores from different layers on a common scale. With a single global threshold and *no retraining*, AIRE-PRUNE achieves strong compression/accuracy trade-offs across the LRA benchmark.

**Summary of findings.** Averaged over tasks for S5, AIRE-PRUNE removes **60.8%** of states while incurring only **0.29** percentage points of accuracy loss. Under a $\leq 1$ pp loss budget, it safely prunes **80%** on *Text* and *Pathfinder*, **70%** on *Path-X*, **65%** on *Image*, **50%** on *Retrieval*, and even **20%** on the often-labeled "incompressible" *ListOps*. Across the board, AIRE-Prune dominates across all the baselines.

**Practical implications.** Because scores are computed once and applied globally, practitioners can: (i) set a target budget without per-layer tuning; (ii) sweep to the elbow and stop just before it to maximize savings at fixed accuracy; and (iii) realize system-level gains by deleting entire layers whenever their states fall below threshold, converting fine-grained sparsity into *block-structured* reductions that are easier to deploy on hardware.

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

# A APPENDIX

## A.1 MATHEMATICAL PROOFS

### A.1.1 VECTOR/MATRIX INNER PRODUCTS AND NORMS

**Euclidean space.** For $\boldsymbol{x} \in \mathbb{C}^n$, the standard inner product and induced 2-norm are

$$\langle \boldsymbol{x}, \boldsymbol{y} \rangle := \boldsymbol{x}^* \boldsymbol{y}, \qquad \|\boldsymbol{x}\|_2 := \sqrt{\boldsymbol{x}^* \boldsymbol{x}}.$$

The Cauchy–Schwarz inequality gives $|\langle \boldsymbol{x}, \boldsymbol{y} \rangle| \leq \|\boldsymbol{x}\|_2 \|\boldsymbol{y}\|_2$.

**Frobenius (Hilbert–Schmidt) structure.** For $\boldsymbol{A}, \boldsymbol{B} \in \mathbb{C}^{m \times n}$, the Frobenius inner product and norm are

$$\langle \boldsymbol{A}, \boldsymbol{B} \rangle_F := \operatorname{tr}(\boldsymbol{A}^* \boldsymbol{B}), \qquad \|\boldsymbol{A}\|_F := \sqrt{\operatorname{tr}(\boldsymbol{A}^* \boldsymbol{A})} = \left( \sum_{i,j} |a_{ij}|^2 \right)^{1/2}.$$

This norm is: (i) *unitarily invariant* ( $\|\boldsymbol{U}\boldsymbol{A}\boldsymbol{V}\|_F = \|\boldsymbol{A}\|_F$ for unitary $\boldsymbol{U}, \boldsymbol{V}$ ), (ii) *compatible* with vectorization ( $\|\boldsymbol{A}\|_F = \|\operatorname{vec}(\boldsymbol{A})\|_2$ ), and (iii) *submultiplicative* with respect to the spectral norm: $\|\boldsymbol{A}\boldsymbol{B}\|_F \leq \|\boldsymbol{A}\|_2 \|\boldsymbol{B}\|_F$ and $\|\boldsymbol{A}\boldsymbol{B}\|_F \leq \|\boldsymbol{A}\|_F \|\boldsymbol{B}\|_2$.

**Rank-1 outer products.** Given $\boldsymbol{u} \in \mathbb{C}^m$ and $\boldsymbol{v} \in \mathbb{C}^n$, the outer product $\boldsymbol{u}\boldsymbol{v}^* \in \mathbb{C}^{m \times n}$ has entries $(\boldsymbol{u}\boldsymbol{v}^*)_{ij} = u_i \overline{v_j}$.

### A.1.2 IDENTITY FOR RANK-1 TERMS

**Lemma 1** (Frobenius norm of a rank-1 outer product). *For $\boldsymbol{u} \in \mathbb{C}^m$ and $\boldsymbol{v} \in \mathbb{C}^n$,*

$$\|\boldsymbol{u}\boldsymbol{v}^*\|_F^2 = \|\boldsymbol{u}\|_2^2 \|\boldsymbol{v}\|_2^2.$$

*Proof.*

$$\|\boldsymbol{u}\boldsymbol{v}^*\|_F^2 = \sum_{i,j} |u_i \overline{v_j}|^2 = \left( \sum_i |u_i|^2 \right) \left( \sum_j |v_j|^2 \right) = \|\boldsymbol{u}\|_2^2 \|\boldsymbol{v}\|_2^2.$$

$\square$

### A.1.3 "ENERGY" INTERPRETATION OF $\|\boldsymbol{H}_t\|_F^2$

For an LTI system with $\boldsymbol{y}_t = \sum_{k \geq 0} \boldsymbol{H}_k \boldsymbol{u}_{t-k}$ and a zero-mean white input $\boldsymbol{u}_t \sim \mathcal{CN}(0, \boldsymbol{I}_h)$ independent across $t$, the instantaneous output power contributed by the $k$-lag kernel is

$$\mathbb{E} \|\boldsymbol{H}_k \boldsymbol{u}_{t-k}\|_2^2 = \mathbb{E} \operatorname{tr}(\boldsymbol{u}_{t-k}^* \boldsymbol{H}_k^* \boldsymbol{H}_k \boldsymbol{u}_{t-k}) = \operatorname{tr}(\boldsymbol{H}_k^* \boldsymbol{H}_k \mathbb{E}[\boldsymbol{u}_{t-k} \boldsymbol{u}_{t-k}^*]) = \operatorname{tr}(\boldsymbol{H}_k^* \boldsymbol{H}_k) = \|\boldsymbol{H}_k\|_F^2.$$

Thus, $\|\boldsymbol{H}_k\|_F^2$ equals the expected output energy (power for unit-variance inputs) contributed by the $k$-th impulse slice. The same holds componentwise for $\boldsymbol{H}_t^{(i)}$, so $\|\boldsymbol{H}_t^{(i)}\|_F^2$ quantifies the energy carried by the $i$-th mode at lag $t$. Moreover, since $\|\boldsymbol{H}_t\|_F^2 = \sum_i \|\boldsymbol{H}_t^{(i)}\|_F^2$ when the rank-1 terms are mutually orthogonal in the Frobenius inner product, or more generally $\|\boldsymbol{H}_t\|_F^2 = \operatorname{tr}(\boldsymbol{H}_t^* \boldsymbol{H}_t)$ always, this norm provides a natural, additive energy accounting across modes and lags.

### A.1.4 CONNECTION TO THE $\mathcal{H}_2$ NORM (TOTAL ENERGY)

For a stable MIMO LTI system with transfer matrix $\boldsymbol{G}(z)$ and impulse sequence $\{\boldsymbol{H}_t\}$,

$$\|\boldsymbol{G}\|_{\mathcal{H}_2}^2 = \frac{1}{2\pi} \int_0^{2\pi} \operatorname{tr}(\boldsymbol{G}(e^{j\omega})^* \boldsymbol{G}(e^{j\omega})) \, d\omega = \sum_{t=0}^{\infty} \|\boldsymbol{H}_t\|_F^2,$$

by Parseval/Plancherel (discrete-time). Hence the Frobenius-squared of impulse slices sums to the *total* output energy for white inputs. This makes $\|\boldsymbol{H}_t\|_F^2$ a natural energy density over lags.

A.1.5 REAL VS. COMPLEX DATA; BIDIRECTIONALITY

If the model uses complex pairs to represent real-valued dynamics, terms appear as conjugate pairs whose sum is real. Use $\boldsymbol{v}^*$ (not $\boldsymbol{v}^\top$) in complex algebra; in purely real settings, $\boldsymbol{v}^* = \boldsymbol{v}^\top$. For bidirectional SSMs, one can form an augmented LTI with block-diagonal forward/backward dynamics; the energy accounting above applies componentwise and adds.

## B FROM ENERGY-BASED PRUNING TO WORST-CASE CERTIFICATES

### B.1 SCOPE AND PROMISE (WHAT THIS APPENDIX DELIVERS)

- **What AIRE-Prune proposes.** An *energy-based*, post-training, layer-adaptive pruning rule for diagonal (or diagonalizable) state-space layers that uses a single global threshold.
- **What we add here.** A precise, *worst-case* ($H_\infty$) *error certificate* that complements the original *typical-case* (energy/$H_2$) rationale, and an end-to-end distortion bound for a residual stack with Lipschitz wrappers.
- **Why it matters.** You keep the same AIRE ranking/thresholding, but now with a computable, auditable guarantee; we also position the result against LAST (Gwak et al., 2025) and highlight the mathematical differences.

### B.2 BACKGROUND: DIAGONAL SSM LAYERS AND ENERGY (NO PRIOR KNOWLEDGE ASSUMED)

**Diagonal/diagonalized SSM layer.** Let $\boldsymbol{x}_{k+1} = \boldsymbol{\Lambda}\boldsymbol{x}_k + \boldsymbol{B}\boldsymbol{u}_k$ and $\boldsymbol{y}_k = \boldsymbol{C}\boldsymbol{x}_k$, with $\boldsymbol{\Lambda} = \mathrm{diag}(\lambda_1, \ldots, \lambda_n), |\lambda_i| < 1$ for stability. The impulse response and frequency response are

$$\boldsymbol{H}_t = \boldsymbol{C}\boldsymbol{\Lambda}^t\boldsymbol{B} \quad (t = 0, 1, 2, \ldots), \qquad \boldsymbol{G}(e^{j\omega}) = \boldsymbol{C}(\boldsymbol{I} - \boldsymbol{\Lambda}e^{-j\omega})^{-1}\boldsymbol{B} = \sum_{i=1}^{n} \frac{\boldsymbol{C}_{:,i}\boldsymbol{B}_{i,:}}{1 - \lambda_i e^{-j\omega}}.$$

Each mode $i$ contributes a rank-1 impulse slice $\boldsymbol{H}_t^{(i)} = \boldsymbol{C}_{:,i}\lambda_i^t\boldsymbol{B}_{i,:}$.

**Energy ($H_2$) of a layer and a mode.** Parseval implies

$$\|\Sigma\|_{\text{energy}}^2 = \sum_{t=0}^{\infty} \|\boldsymbol{H}_t\|_F^2 = \frac{1}{2\pi} \int_0^{2\pi} \|\boldsymbol{G}(e^{j\omega})\|_F^2 \, d\omega.$$

Using $\|\boldsymbol{u}\boldsymbol{v}^*\|_F^2 = \|\boldsymbol{u}\|_2^2\|\boldsymbol{v}\|_2^2$ and the geometric series,

$$E_i \triangleq \sum_{t=0}^{\infty} \|\boldsymbol{H}_t^{(i)}\|_F^2 = \frac{\|\boldsymbol{C}_{:,i}\|_2^2 \|\boldsymbol{B}_{i,:}\|_2^2}{1 - |\lambda_i|^2}, \qquad \|\Sigma\|_{\text{energy}}^2 = \sum_{i=1}^{n} E_i.$$

*Intuition.* $E_i$ is the mode's long-run output power (e.g., under unit white-noise inputs). If $\boldsymbol{B}$ rows are normalized, then $E_i = \|\boldsymbol{C}_{:,i}\|_2^2/(1 - |\lambda_i|^2)$.

### B.3 AIRE-PRUNE: THE ALGORITHM

1. **Per-mode energy.** Compute $E_i$ for each state in each layer.
2. **Sort & prefix sums (per layer).** Sort $E_{(1)} \geq \cdots \geq E_{(n)}$; set $S(i) = \sum_{j \leq i} E_{(j)}$.
3. **Prefix-normalized score.**
$$s(i) = \frac{E_{(i)}}{S(i) + \varepsilon}, \qquad 0 < \varepsilon \ll 1.$$
   This score is monotone non-increasing in $i$; a single global threshold $\tau$ causes each layer to *keep its longest prefix* and *prune a contiguous tail*.
4. **Global threshold.** Concatenate all $s(i)$ across layers, choose $\tau$ for the desired global budget, and materialize kept/pruned indices per layer.

*Practical effect.* The prefix rule yields clear head-tail separation and, at higher thresholds, may delete entire low-energy layers, which helps wall-clock latency.

### B.4 WHY DISCUSS WORST-CASE BOUNDS IF ENERGY ALREADY WORKS?

The energy lens ($H_2$) explains typical/average behavior and aligns with many workloads. We therefore derive an $H_\infty$ *certificate from* the same energy quantities AIRE already computes, plus a single stability factor. The method remains unchanged; the mathematics provides assurance why AIRE-Prune works.

#### WHY THE BOUND SHOULD HOLD

Sort a layer's states by energy. Cumulative energy vs. kept states shows an elbow: most energy sits in a short head. Pruning the tiny tail barely changes the typical output. Our result shows the *worst-case* change is also small unless pruned poles are extremely close to the unit circle. The stability margin enters through a single factor.

### B.5 PER-LAYER WORST-CASE ($H_\infty$) CERTIFICATE FROM ENERGY TAILS

**Goal.** We want to bound the worst-case (induced $\ell_2 \to \ell_2$) error of a layer after pruning a set of modes $\mathcal{T}$:

$$\varepsilon \triangleq \|\boldsymbol{G} - \widetilde{\boldsymbol{G}}\|_\infty = \sup_{\omega \in [0, 2\pi)} \Big\| \sum_{i \in \mathcal{T}} \boldsymbol{G}_i(e^{j\omega}) \Big\|_2, \qquad \boldsymbol{G}_i(e^{j\omega}) = \frac{\boldsymbol{C}_{:,i}\boldsymbol{B}_{i,:}}{1 - \lambda_i e^{-j\omega}}.$$

**Step 1: Triangle inequality under the $H_\infty$ norm.** The induced (operator) 2-norm satisfies $\|\boldsymbol{A} + \boldsymbol{B}\|_2 \le \|\boldsymbol{A}\|_2 + \|\boldsymbol{B}\|_2$. Taking the supremum over $\omega$ preserves the inequality:

$$\varepsilon = \sup_\omega \Big\| \sum_{i \in \mathcal{T}} \boldsymbol{G}_i(e^{j\omega}) \Big\|_2 \overset{\text{(triangle)}}{\le} \sup_\omega \sum_{i \in \mathcal{T}} \|\boldsymbol{G}_i(e^{j\omega})\|_2$$

$$\overset{\text{(sup subadditivity)}}{\le} \sum_{i \in \mathcal{T}} \sup_\omega \|\boldsymbol{G}_i(e^{j\omega})\|_2 = \sum_{i \in \mathcal{T}} \|\boldsymbol{G}_i\|_\infty.$$

Hence,

$$\varepsilon \le \sum_{i \in \mathcal{T}} \|\boldsymbol{G}_i\|_\infty. \tag{12}$$

**Step 2: Per-mode $H_\infty$ envelope.** Write $\boldsymbol{G}_i(e^{j\omega}) = \dfrac{\boldsymbol{C}_{:,i}\boldsymbol{B}_{i,:}}{1 - \lambda_i e^{-j\omega}}$. For a rank-1 matrix $\boldsymbol{uv}^*$, $\|\boldsymbol{uv}^*\|_2 = \|\boldsymbol{u}\|_2 \|\boldsymbol{v}\|_2$, so define

$$\alpha_i \triangleq \|\boldsymbol{C}_{:,i}\|_2 \|\boldsymbol{B}_{i,:}\|_2 = \|\boldsymbol{C}_{:,i}\boldsymbol{B}_{i,:}\|_2.$$

For any $\omega$, the reverse triangle inequality gives $|1 - \lambda_i e^{-j\omega}| \ge 1 - |\lambda_i|$. Therefore,

$$\|\boldsymbol{G}_i(e^{j\omega})\|_2 = \frac{\alpha_i}{|1 - \lambda_i e^{-j\omega}|} \le \frac{\alpha_i}{1 - |\lambda_i|} \quad \Rightarrow \quad \|\boldsymbol{G}_i\|_\infty \le \frac{\alpha_i}{1 - |\lambda_i|}. \tag{13}$$

**Step 3: Express the peak envelope via the energy $E_i$.** By definition of the per-mode energy (asymptotic impulse/white-noise energy),

$$E_i = \frac{\|\boldsymbol{C}_{:,i}\|_2^2 \|\boldsymbol{B}_{i,:}\|_2^2}{1 - |\lambda_i|^2} = \frac{\alpha_i^2}{1 - |\lambda_i|^2}.$$

Solving for $\alpha_i$ and substituting into equation 13:

$$\alpha_i = \sqrt{E_i} \sqrt{1 - |\lambda_i|^2} \quad \Rightarrow \quad \frac{\alpha_i}{1 - |\lambda_i|} = \sqrt{E_i} \frac{\sqrt{1 - |\lambda_i|^2}}{1 - |\lambda_i|} = \sqrt{E_i} \sqrt{\frac{1 + |\lambda_i|}{1 - |\lambda_i|}}.$$

Thus, for every pruned mode $i$,

$$\|\boldsymbol{G}_i\|_\infty \le \sqrt{E_i} \sqrt{\frac{1 + |\lambda_i|}{1 - |\lambda_i|}}. \tag{14}$$

**Step 4: Uniform "stability factor" over the pruned set.** Let

$$\rho \triangleq \max_{i \in \mathcal{T}} |\lambda_i| \quad \text{and} \quad \kappa(\rho) \triangleq \sqrt{\frac{1+\rho}{1-\rho}}.$$

The scalar function $x \mapsto \sqrt{\frac{1+x}{1-x}}$ is increasing on $[0,1)$, so for all $i \in \mathcal{T}$,

$$\sqrt{\frac{1+|\lambda_i|}{1-|\lambda_i|}} \leq \kappa(\rho).$$

Combining with equation 14 yields the uniform bound

$$\|\boldsymbol{G}_i\|_\infty \leq \kappa(\rho)\sqrt{E_i}, \qquad \forall i \in \mathcal{T}. \tag{15}$$

**Step 5: First aggregate bound (linear in $\sqrt{E_i}$).** Substitute equation 15 into equation 12:

$$\varepsilon \leq \sum_{i \in \mathcal{T}} \kappa(\rho)\sqrt{E_i} = \kappa(\rho)\sum_{i \in \mathcal{T}}\sqrt{E_i}. \tag{16}$$

**Step 6: Second aggregate bound via Cauchy–Schwarz (root-of-sum).** For nonnegative $\{a_i\}$ and $m = |\mathcal{T}|$, Cauchy–Schwarz gives $\sum_{i=1}^m a_i \leq \sqrt{m}\sqrt{\sum_{i=1}^m a_i^2}$. Apply this with $a_i = \sqrt{E_i}$:

$$\sum_{i \in \mathcal{T}}\sqrt{E_i} \leq \sqrt{|\mathcal{T}|}\sqrt{\sum_{i \in \mathcal{T}} E_i}.$$

Combining with equation 16 yields

$$\varepsilon \leq \kappa(\rho)\sqrt{|\mathcal{T}|}\sqrt{\sum_{i \in \mathcal{T}} E_i}. \tag{17}$$

**Step 7: Final certificate (take the minimum).** Both equation 16 and equation 17 are valid; each can be tighter depending on whether the pruned tail is concentrated or diffuse. We therefore report the minimum:

$$\boxed{\varepsilon = \|\boldsymbol{G} - \widetilde{\boldsymbol{G}}\|_\infty \leq \kappa(\rho)\min\left\{\sum_{i \in \mathcal{T}}\sqrt{E_i}, \sqrt{|\mathcal{T}|}\sqrt{\sum_{i \in \mathcal{T}} E_i}\right\}.}$$

**Why the bound is finite (stability).** Since the layer is stable, $|\lambda_i| < 1$ for all $i$, so each $E_i = \alpha_i^2/(1-|\lambda_i|^2)$ is finite and $\rho < 1$, which implies $\kappa(\rho) = \sqrt{(1+\rho)/(1-\rho)} < \infty$.

**Special cases.** If $\boldsymbol{B}$ rows are normalized, then $E_i = \|\boldsymbol{C}_{:,i}\|_2^2/(1-|\lambda_i|^2)$ (the derivation above applies verbatim). For bidirectional layers, sum (or average) the forward/backward $\|\boldsymbol{C}\|_2^2$ contributions when forming $E_i$; the rank-1 and norm inequalities used here are unchanged.

## B.6 Mathematical comparison to LAST (and theory positioning)

**Per-mode quantities (shorthand).** $\alpha_i \triangleq \|\boldsymbol{C}_{:,i}\|_2\|\boldsymbol{B}_{i,:}\|_2$, $E_i = \dfrac{\alpha_i^2}{1-|\lambda_i|^2}$, $g_i^{(\infty)} \leq$

$\dfrac{\alpha_i}{1-|\lambda_i|}$, $\rho = \max_{i \in \mathcal{T}}|\lambda_i|$, $\kappa(\rho) = \sqrt{\dfrac{1+\rho}{1-\rho}}$.

**Layer-level envelopes (apples-to-apples).**

$$\textbf{LAST (H}\infty\textbf{-first)}: \qquad \|\boldsymbol{G} - \widetilde{\boldsymbol{G}}\|_\infty \lesssim \sum_{i \in \mathcal{T}}\frac{\alpha_i}{1-|\lambda_i|} \tag{18}$$

$$\textbf{AIRE (energy-first} \to \textbf{H}\infty): \qquad \|\boldsymbol{G} - \widetilde{\boldsymbol{G}}\|_\infty \leq \kappa(\rho)\min\left\{\sum_{i \in \mathcal{T}}\frac{\alpha_i}{\sqrt{1-|\lambda_i|^2}}, \sqrt{|\mathcal{T}|}\sqrt{\sum_{i \in \mathcal{T}}\frac{\alpha_i^2}{1-|\lambda_i|^2}}\right\}.$$
$$\tag{19}$$

**Key mathematical differences.**

1. *Aggregation.* LAST sums *peak gains* linearly, $\sum \alpha_i / (1 - |\lambda_i|)$. AIRE aggregates *energies* either linearly in $\sqrt{E_i}$ or sublinearly via $\sqrt{|\mathcal{T}| \sum E_i}$, which is strictly tighter for *diffuse* tails (Cauchy–Schwarz gap).

2. *Pole-radius dependence.* As $\rho \uparrow 1$, $1 - |\lambda|^2 \approx 2(1 - |\lambda|)$. Both equation 18 and the first term in equation 19 scale like $(1 - \rho)^{-1}$; AIRE's $\kappa(\rho)$ makes this dependence explicit and uniform across the pruned set.

3. *Quantities needed.* LAST needs $\alpha_i$ and $(1 - |\lambda_i|)^{-1}$ per pruned mode. AIRE needs $E_i$ (already computed for ranking) and a single $\rho$. No new per-mode statistics are required beyond AIRE's pipeline.

4. *Cross-layer coupling.* LAST's guarantee is tied to per-mode peak surrogates. AIRE's certificate is *tail-centric* (energy tail + one $\rho$), aligning with prefix-normalized, contiguous-tail pruning and explaining whole-layer drops.

**When AIRE can be tighter.** *Diffuse tails:* if $|\mathcal{T}|$ is large but $\sum E_i$ is tiny, then

$$\kappa(\rho) \sqrt{|\mathcal{T}| \sum E_i} \ll \sum_{i \in \mathcal{T}} \frac{\alpha_i}{1 - |\lambda_i|},$$

making AIRE's root-of-sum form substantially less conservative.

**Conceptual positioning.** LAST is worst-case-first and can be conservative when worst frequencies are rarely excited. AIRE is typical-case-first (energy/$H_2$), empirically enabling larger safe pruning.

## C  EXPERIMENTAL DETAILS

All S5 experiments were implemented in JAX (Bradbury et al., 2018) and run on a single NVIDIA A100 / H100 accelerator (40 GB or 80 GB VRAM). Our models and training code are based on the public S5 (Smith et al., 2023)[1], S4D (Gu et al., 2022a)[2] and Mamba(S6) Gu & Dao, 2023[3] implementation. Unless otherwise stated, we employ bidirectional SSM layers for all LRA tasks, and use the parallel-scan inference kernel provided by S5.

### C.1  BENCHMARKS / TASKS

We evaluate on the Long Range Arena (LRA), a suite of long-context sequence problems spanning symbolic reasoning, byte-level text, document retrieval, and flattened vision, designed to probe modeling of dependencies across sequences up to 16k tokens.

**ListOps** 10-way classification over extended ListOps expressions (Nangia & Bowman, 2018). Inputs are single-channel sequences (max length 2,048) encoding digits, operators, and bracket markers as one-hot vectors over 17 tokens. Splits: 96k train, 2k validation, 2k test.

**Text** Binary sentiment classification on IMDB reviews at the byte level (Maas et al., 2011). Each example is a single-channel sequence up to 4,096 tokens, using a 129-symbol one-hot alphabet. Splits: 25k train, 25k test.

**Retrieval** Binary document-pair classification on the ACL Anthology Network (Radev et al., 2009). The goal is to predict whether two documents share equivalent citation links. Each document is byte-tokenized with a 97-symbol one-hot encoding and capped at 4,000 tokens. Splits: 147,086 train pairs, 18,090 validation pairs, 17,437 test pairs.

**Image** 10-way classification on flattened CIFAR-10 (Krizhevsky & Hinton, 2009), represented as single-channel sequences of length 1,024.

---

[1] https://github.com/lindermanlab/S5
[2] https://github.com/srush/annotated-s4
[3] https://github.com/jsie7/ssm-benchmark

Table 4: Training configurations for S5 models on the six LRA tasks. All runs use batch normalization, pre-normalization, and $\Delta_{\max} = 0.1$. $n_m$: SSM state dimension; $J$: blocks for diagonal/block init of $\Lambda$; $D$: dropout; LR: global learning rate; SSM LR: learning rate for SSM-only params; $B$: batch size; $E$: epochs; WD: weight decay; $\Delta_{\min}$: minimum step.

| Task | L | h | $n_m$ | J | D | LR | SSM LR | B | E | WD | $\Delta_{\min}$ |
|------|---|---|-------|---|---|-----|--------|---|---|-----|------------|
| ListOps | 8 | 128 | 16 | 8 | 0 | 0.003 | 0.001 | 50 | 40 | 0.07 | 0.001 |
| Text | 6 | 256 | 192 | 12 | 0.1 | 0.004 | 0.001 | 50 | 35 | 0.07 | 0.001 |
| Retrieval | 6 | 128 | 256 | 16 | 0 | 0.002 | 0.001 | 32 | 20 | 0.05 | 0.001 |
| Image | 6 | 512 | 384 | 3 | 0.1 | 0.005 | 0.001 | 50 | 250 | 0.07 | 0.001 |
| Pathfinder | 6 | 192 | 256 | 8 | 0.05 | 0.005 | 0.0009 | 64 | 200 | 0.07 | 0.001 |
| Path-X | 6 | 128 | 256 | 16 | 0 | 0.002 | 0.0006 | 32 | 75 | 0.05 | 0.001 |

Table 5: AIRE-Prune: pruning ratios and resulting inference speedups and parameter-memory reductions for **Mamba (S6)**

| Metric ↓ / Dataset → | ListOps | Text | Retrieval | Image | Pathfinder | Path-X |
|----------------------|---------|------|-----------|-------|------------|--------|
| **Pruning ratio** | – | 45% | 60% | 80% | – | – |
| **Inference speedup ($\times$)** | – | $1.61\times$ | $1.57\times$ | $2.03\times$ | – | – |
| **Parameter reduction (%)** | – | 19.1% | 8.1% | 12.1% | – | – |

**Pathfinder** Binary classification on flattened Pathfinder stimuli (Linsley et al., 2018), determining whether two points are connected by a target path among distractors. Sequences are single-channel with length 1,024. Splits: 160k train, 20k validation, 20k test.

**Path-X** A scaled, more challenging Pathfinder variant (Linsley et al., 2018) with single-channel sequences of length 16,384, again testing path connectivity under heavy clutter.

**Speech** Speech command is a 35-way classification task on 1-second spoken utterances (Warden, 2018). Each audio clip is a single-channel waveform of length 16,000 samples (16 kHz). For experiments with different sampling rates, we additionally create a downsampled version at 8 kHz. The resulting split contains 24,482 training examples, 5,246 validation examples, and 5,247 test examples.

## C.2 HYPERPARAMETER

We follow the LRA protocol across six tasks—ListOps, Text, Retrieval, Image, Pathfinder, and Path-X—tuning depth ($L$), hidden width ($h$), and SSM state size ($n_m$) per task. Byte-level Text and long-context Retrieval favor larger $J$ (more blocks) and higher $n_m$ to capture long-range dependencies; flattened Image/Pathfinder variants use wider channels ($h$) with moderate dropout $D$ for regularization. Learning rates are decoupled, with a smaller SSM LR than the global LR to stabilize eigen-parameter updates, and modest weight decay (WD) throughout. Batch sizes $B$ and epochs $E$ reflect dataset scale (e.g., longer training for Image/Pathfinder), while $\Delta_{\min}$ fixes the minimum step size used in state discretization. This setup (as in Table 4 provides a consistent, comparable training recipe across heterogeneous long-context workloads.

## C.3 EFFICIENCY IMPACT OF DIMENSION-REDUCED S5 MODELS

To assess the practical efficiency impact of AIRE-Prune for mamba (S6), we implemented pruning by removal in addition to the standard masking-based implementation, transferring the retained high-importance states into a lower-dimensional S5 model. Table 5 reports the average evaluation-step throughput and the corresponding parameter reduction achieved by the pruned S5 models on an NVIDIA H100 GPU. Shrinking the state dimension consistently improves both compute cost and model size, with the magnitude of savings varying across tasks as a function of channel width.

