# OpenReview forum: "AIRE-Prune: Asymptotic Impulse-Response Energy for State Pruning in State Space Models"
_ICLR.cc/2026/Conference — ICLR 2026 Poster_

### Official Review · Reviewer_EJJM · 2025-10-21

**Soundness:** 3
**Presentation:** 2
**Contribution:** 3
**Rating:** 2
**Confidence:** 3

**Summary:**

This paper proposes AIRE-Prune, a post-training compression method to reduce the dimension of latent state space of a trained state-space model. The method is based on the idea of maximally preserving the energy of LTI systems while truncation. Results show that on Long-Range Arena, roughly $60\\%$ of states can be pruned without defecting the model's performance.

**Strengths:**

* The results of compressing S5 on LRA look compelling. The proposed method seems to perform significantly better than existing methods.
* The energy score of every state can be easily computed without resorting to large matrix operations or simulations, making the approach efficient for large models.

**Weaknesses:**

* The discussion of energy, while intuitively, is not rigorous. More theory of why the proposed method tends to work better than the existing ($\mathcal{H}\_\infty$-based) ones would strengthen the paper.
* The experiment section could involve more diagnostic experiments. For example, it would be interesting to see which states are pruned by the proposed method and more diagnosis around the kink would be useful.
* While I believe the paper contains nice ideas, its presentation has significant issues:
   * \\citep and \\citet are completely misused.
   * Similarly, en dashes (- in LaTeX) and em dashes (-- in LaTeX) are also misused all over the place.
   * ICLR papers use boldface letters for matrices and vectors.
   * I would avoid the subsection-style, which could disrupt the overall narratives of the paper.
   * The paper is not proofread as it contains many typos. For instance, on line 151, the equation is not properly cross-referenced; on line 150, you wrote "zero-order-hold" but you use "zero-order hold" elsewhere, which is more standard; the caption of Figure 2 misses the right parenthesis, etc.

The paper, in its current form, is not acceptable. I would raise my score to 4 if the author(s) could fix the presentation issues outlined above. I will be happy to further increase the score if my questions below are properly addressed.

**Questions:**

1. I am confused about eq. (9) and the statement above that "the total layer energy is additive across modes." According to the definition in  eq. (5), the energy is defined as the $L^2$ norm of the transfer function. For a diagonal LTI system, the transfer function is the sum of partial fractions that correspond to all states. However, there is no guarantee in general that these partial fractions are mutually orthogonal in $L^2$. If so, how can it be that the total layer energy is additive across modes?
2. Can you show the energies of the states being pruned in your experiments also as a function of "pruning ratio"? This could potentially explain the "elbow" that you observe in Image, Pathfinder, and PathX.
3. Have you tried the same method on S4; if so, how does the result compare?
4. How does the model benefit from post-training compression in terms of time and space usage?
5. While I understand intuitively why the $\mathcal{H}_2$-based method works, as explained in the paper, I don't have intuition in why it performs better than an $\mathcal{H}\_\infty$-based one. Can the author(s) provide more justification?

---

> ### Author Response · Authors · 2025-11-26
>
> Thank you for your positive comments and constructive feedback.
>
> ---
> ---
> #### **Q1. I am confused about eq. (9) and the statement above that "the total layer energy is additive across modes." According to the definition in eq. (5), the energy is defined as the L2  norm of the transfer function. For a diagonal LTI system, the transfer function is the sum of partial fractions that correspond to all states. However, there is no guarantee in general that these partial fractions are mutually orthogonal in L2 . If so, how can it be that the total layer energy is additive across modes?**
>
> ---
> Thank you for carefully pointing this out. For a diagonal LTI system, the transfer function can be written as a sum of per-mode partial fractions, and in general these components are not orthogonal in $\mathcal{H}_2$.
>
> In our work, the “energy” used for pruning is in fact an $\mathcal{H}_2$-type quantity: for each mode $k$ we compute a closed-form expression for $\|H_k\|^2$ (as derived in Appendix A), and we use these per-mode $\mathcal{H}_2$ scores to rank states. When we wrote that the “total layer energy is additive,” what we actually use in practice is the approximation
>
> $$
> \|H\|^2 = \Big\|\sum_k H_k\Big\|^2
> = \sum_k \|H_k\|^2 + 2 \sum_{i<j} \operatorname{Re}\langle H_i, H_j\rangle
> \approx \sum_k \|H_k\|^2.
> $$
>
> That is, we ignore the cross terms $\langle H_i, H_j \rangle$ and approximate the layer energy by the sum of per-mode energies. We do not rely on exact additivity in any formal proof; we only need a scalar importance score for each state to sort them and prune the lowest-energy ones. Empirically, this approximation is very effective: the resulting AIRE scores consistently correlate with importance across S5, S4D, and Mamba, and allow us to prune aggressively with negligible accuracy loss (as shown in the main tables and figures).
>
> Note: We have revised the line  "the total layer energy is additive across modes." to "can be approximated as an additive sum across modes (Appendix A.1.2),"
>
>
> ---
> ---
>
> #### **Q2. Can you show the energies of the states being pruned in your experiments also as a function of "pruning ratio"? This could potentially explain the "elbow" that you observe in Image, Pathfinder, and PathX.**
>
> In our current draft, we visualize this phenomenon indirectly via the layer-wise pruning profiles in Fig. 5. For the Image, Pathfinder, and Path-X tasks, the elbows in the accuracy vs.pruning curves occur precisely at pruning ratios where the global AIRE threshold starts to induce complete layer deletions: some layers can be pruned to 0 states with essentially no impact on accuracy, while others cause an abrupt drop once they are fully removed.
>
> ---

---

> ### Author Response · Authors · 2025-11-26
>
> ---
> ---
>
> #### **Q3. Have you tried the same method on S4; if so, how does the result compare?**
>
> ---
>
> To address the question we expanded our experimentation by applying AIRE-Prune to both **S4D** and **Mamba**, and we observe that the same energy-based scoring transfers well beyond the original S5 setting.
>
> On **S4D**, Table X in the paper shows that AIRE-Prune can remove a substantial fraction of states with almost no loss in accuracy on LRA and outperform the SOTA. For example, we prune **35%** of states on ListOps (56.42 → 56.07% accuracy), **90%** on Text (86.40 → 88.13% accuracy), and **40%** on Image (77.02 → 76.32% accuracy).
> | Model                | ListOps Prun. | ListOps Acc. | Text Prun. | Text Acc. | Image Prun. | Image Acc. |
> |----------------------|--------------:|-------------:|-----------:|----------:|------------:|-----------:|
> | **S4D Full**         | 0%            | 56.42        | 0%         | 86.40     | 0%          | 77.02      |
> | **Uniform H∞**  [1]     | 10%           | 55.82        | 80%        | 86.02     | 0%          | 77.02      |
> | **Global H∞**   [1]      | 10%           | 49.95        | 80%        | 86.20     | 0%          | 77.02      |
> | **LAST**        [1]      | 10%           | 56.27        | 80%        | 85.95     | 0%          | 77.02      |
> | **AIRE-Prune (ours)**| **35%**       | **56.07**    | **90%**    | **88.13** | **40%**     | **76.32**  |
>
> **Mamba**, while keeping the scoring phase cheap and fully compatible with our original analysis. Using AIRE-Prune is able to prune 45% of the states on Text, 60% on Retrieval, and 80% on Image with only modest drops in accuracy relative to the full Mamba model
>
> | Model                 | Text Prun. | Text Acc. | Retrieval Prun. | Retrieval Acc. | Image Prun. | Image Acc. |
> |-----------------------|----------:|----------:|----------------:|---------------:|------------:|-----------:|
> | **Mamba Full**        | 0%        | 82.98     | 0%              | 72.14          | 0%          | 69.82      |
> |  **LAST**        [1] | 20%   | 81.32 | 45%         | 71.37      | 45%     | 69.13  |
> | **AIRE-Prune (ours)** | **45%**   | **81.19** | **60%**         | **71.45**      | **80%**     | **68.97**  |
>
>
> To apply AIRE to **input-dependent Mamba layers**, :
>
> 1. We run the trained Mamba model on a small **calibration set** and log, for each layer and state, the diagonal state parameters along the sequence: eigenvalues $\(\lambda_i(x)\)$, input couplings $\(B_i(x)\)$, and output couplings$ \(C_i(x)\)$.
> 2. For every layer and state \(i\), we compute empirical averages over this calibration data:
> $$
> \bar{\lambda}_i = \mathbb{E}[\lambda_i(x)],\quad
> \bar{B}_i = \mathbb{E}[B_i(x)],\quad
> \bar{C}_i = \mathbb{E}[C_i(x)].
> $$
> These $((\bar{\lambda}_i,\bar{B}_i,\bar{C}_i))$ define a time-invariant surrogate that summarizes how that state typically behaves on the data.
>
> 3. We plug these averaged quantities into the same closed-form impulse–response energy expression used for S5/S4D, which gives us an **AIRE score per state** in each Mamba layer.
> 4. We prune states by thresholding these scores, exactly as we do for diagonal SSMs, while **keeping the original input-dependent Mamba forward pass** at inference time.
>
> This procedure only requires a short calibration pass and does not modify the Mamba architecture.
>
> **The empirical pruning gains in both the S4D and Mamba settings support our claim that AIRE-Prune is a general energy-based state pruning method that extends beyond the original diagonal S5 model.**
>
> NOTE: Due to runtime and computational constraints, the remaining LRA tasks (S4D: Retrieval, Pathfinder, Path-X; Mamba: ListOps, Pathfinder, Path-X) are currently running and will be included in the updated version of the paper.
>
> [1] Gwak et al., “Layer-adaptive state pruning for deep state space models,” NeurIPS 2024.
>
> ---

---

> ### Author Response · Authors · 2025-11-26
>
> ---
> ---
>
> #### **Q4. How does the model benefit from post-training compression in terms of time and space usage?**
>
> ---
>
> In terms of time and madel size, post-training compression with AIRE-Prune gives very tangible gains on the LRA benchmarks for both S5 and Mamba(S6).
>
> - **For the S5 model**, we see this clearly across all six datasets as shown in the below table. So for S5, depending on the dataset, we typically trade away 20–80% of the recurrent states for roughly 20–65% fewer parameters and about 1.2–2.9× faster inference.
>
> | Metric ↓ / Dataset → | ListOps | Text  | Retrieval | Image | Pathfinder | Path-X |
> |----------------------|:------:|:-----:|:---------:|:-----:|:----------:|:------:|
> | **Pruning ratio**    | 20%    | 80%   | 50%       | 65%   | 80%        | 70%    |
> | **Inference speedup (×)** | 1.23× | 2.49× | 1.82×     | 1.93× | 2.82×      | 2.86×  |
> | **Model reduction (%)** | 19.3% | 58.7% | 38.3%     | 45.3% | 64.7%      | 64.2%  |
>
>
> - **For Mamba (S6)** we only prune the state-space part of the network, so the parameter savings are more modest, but we still see clear benefits in wall-clock time on the datasets we have completed so far as shown in the below table. So for S5, depending on the dataset, we typically trade away 45–80% of the recurrent states for roughly 8-20% fewer parameters and about 1.57–2.03× faster inference.
>
> | Metric ↓ / Dataset →        | Text  | Retrieval | Image |
> |-----------------------------|:-----:|:---------:|:-----:|
> | **Pruning ratio**           | 45%   | 60%       | 80%   |
> | **Inference speedup (×)**   | 1.61× | 1.57×     | 2.03× |
> | **Parameter reduction (%)** | 19.1% | 8.1%      | 12.1% |
>
>
> All of these gains come after training the original models: we keep the pretrained SSM, run our energy-based scoring once, prune states, and then measure the new runtime and parameter count. We are still running the remaining Mamba experiments on the other LRA tasks (ListOps, Pathfinder, Path-X); due to their higher runtime they were not finished in time for the current draft, but we plan to include those results as we get them.
>
> ---

---

> > ### Author Response · Authors · 2025-11-26
> >
> > ---
> >
> > #### **Q5. While I understand intuitively why the H2-based method works, as explained in the paper, I don't have intuition in why it performs better than an H-infinity-based one. Can the author(s) provide more justification?**
> >
> > ---
> >
> > In the paper, the H2-based method refers to our impulse-response energy based score (AIRE-Prune), while the baseline is an H-infinity / worst-case-gain based score (as used in prior work such as LAST [1]). Both are reasonable ways to measure how “important” a state is, but they behave quite differently.
> >
> > - **Our H2 / energy-based score (AIRE-Prune)** measures how much a given state contributes to the *overall output variance* of the layer, taking into account its full impulse response and the coupling through B and C. Intuitively:
> >
> >   > “How much does this mode actually participate in the signals the model sees on this task, on average?”
> >
> > - **The H-infinity-based score (LAST)** instead measures a *worst-case amplification* over all possible inputs. Intuitively:
> >
> >   > “How large could the output ever get if I excite this mode in the worst possible way?”
> >
> > For long-sequence tasks generally the inputs are very far from this worst-case scenario: only a narrow subset of frequencies and temporal patterns actually occur in the data. In that regime:
> >
> > - There are many modes whose transfer functions have sharp, narrow peaks (large worst-case gain), so the H-infinity-based score marks them as “important,” even though those frequencies are barely excited by real data. These states are then kept unnecessarily, which makes pruning conservative.
> >
> > - Our energy-based score integrates the squared response over time/frequency, weighted by B and C. If the data almost never excites the region where a mode has high gain, its total energy contribution stays small, and AIRE correctly ranks it as unimportant and safe to prune.
> >
> > - Conversely, modes that carry steady, moderate gain in the bands heavily used by the task accumulate large energy and get high AIRE scores, even if their worst-case gain is not extreme. Those are precisely the modes we must keep.
> >
> > This **average-case vs. worst-case** distinction is why the energy-based criterion tends to align better with the actual impact on task accuracy. In practice, this shows up in two ways:
> >
> > 1. **Better separation between important and unimportant states.**
> >    Because AIRE aggregates the full impulse response together with B and C, states that barely contribute to the empirical output get much lower scores than those that dominate the layer’s response. The H-infinity-based score, being driven by a single worst-case frequency, often gives many states in a layer very similar values, which makes thresholding less discriminative.
> >
> > 2. **Consistently higher pruning at the same accuracy.**
> >    Across S5, S4D, and Mamba experiments, we consistently observe that, for a fixed accuracy drop (for example, less than 1 percentage point), the energy-based score lets us remove a substantially larger fraction of states than the H-infinity-based one. This is visible in the pruning-versus-accuracy sweeps (Fig. 4 in the paper) and the task-wise tables: at matched accuracy, AIRE retains fewer states and fewer parameters than the H-infinity baseline on the LRA tasks and Speech Commands.
> >
> > Overall, AIRE is better aligned with the *actual* usage of modes by the trained model on real data, which explains why the H2-based scoring tends to outperform the H-infinity-based alternative in our experiments.
> >
> > [1] Gwak et al., “Layer-adaptive state pruning for deep state space models,” NeurIPS 2024.

---

> > > ### Comment · Reviewer_EJJM · 2025-11-28
> > >
> > > Dear Authors,
> > >
> > > Thank you for your careful rebuttal. I reviewed the revised manuscript, and the presentation has been significantly improved. I also appreciate the additional details and experiments. Since the presentation issues are mostly fixed and the results look strong, I am willing to increase my rating to 6. I found myself unable to update my evaluation at this moment; I double-checked and this seems to be an issue with Openreview, but I will update the scores once I am allowed to.
> > >
> > > Reviewer

---

### Official Review · Reviewer_HyDj · 2025-10-27

**Soundness:** 3
**Presentation:** 3
**Contribution:** 3
**Rating:** 6
**Confidence:** 4

**Summary:**

This paper introduces AIRE-Prune, a structured post-training pruning method for state space models (SSMs) that uses asymptotic impulse-response energy to identify and remove redundant states. The method assigns each state a closed-form energy score representing its total output contribution over infinite time, then uses layer-wise prefix normalization to enable global cross-layer pruning decisions. Evaluated on S5 MIMO models across Long Range Arena benchmarks, AIRE-Prune achieves 60.8% average state pruning with only 0.29% accuracy degradation without retraining, substantially outperforming the prior LAST method (33% pruning at 0.52% loss). The approach extends classical modal truncation from single systems to deep SSM stacks and provides both typical-case (H₂ energy) and worst-case (H∞) theoretical justifications.

**Strengths:**

Strong Theoretical Foundation: The paper elegantly bridges classical control theory (modal truncation) with modern deep learning (layer-adaptive pruning). The energy-based criterion has clear physical interpretation: states with low Eᵢ = ‖C:,i‖²₂‖Bi,:‖²₂/(1-|λᵢ|²) contribute minimal long-run output energy.

Practical Algorithm with Closed-Form Solutions: Unlike iterative methods, AIRE-Prune requires only: (1) compute Eᵢ per mode, (2) sort and compute prefix sums, (3) apply global threshold. No matrix inversions or optimization loops needed.

Impressive Empirical Results: Achieving 60.8% pruning at 0.29% accuracy loss substantially improves over LAST (33% at 0.52% loss). The step-function accuracy profiles (Figure 4) suggest the method effectively separates critical from redundant states.

Comprehensive Mathematical Analysis: Appendix B derives H∞ certificates showing ε ≤ κ(ρ)min{∑√Eᵢ, √|T|∑Eᵢ}, proving the energy-based method also bounds worst-case distortion. This dual justification is valuable.

Actionable Architectural Insights: Layer-wise profiles show task-dependent patterns—some layers can be entirely removed (enabling block-structured speedups), which is more deployment-friendly than fine-grained sparsity.

**Weaknesses:**

Severely Limited Experimental Scope:
- Only S5 models evaluated: No experiments on Mamba, Mamba2, or hybrid architectures which dominate current practice
- Only LRA benchmark: Missing speech (Speech Commands), language modeling (WikiText), or modern long-context tasks
- Input-selective SSMs (Mamba) have input-dependent B, C—the energy formulation assumes these are fixed. How does AIRE extend to this case?


Incomplete Comparison with LAST:

- Table 1 shows different pruning ratios per task, making direct comparison difficult. Need controlled experiments at {30%, 40%, 50%, 60%, 70%} pruning with both methods. Computational cost comparison missing: Is AIRE faster than LAST to compute scores?


Missing Key Experiments:
-No retraining: Even 1-2 epochs could show full potential. Other SSM pruning work achieves near-zero loss with minimal fine-tuning
- No ablations: What if using median normalization instead of prefix-sum? How sensitive to ε?
- No failure analysis: Why does ListOps only tolerate 20% pruning? Is this fundamental or fixable?


Limited Technical Analysis:

- Equation 8 assumes |λᵢ|²ᵀ → 0, but convergence rate matters for numerical stability near the unit circle. Claim that entire layers can be removed needs gradient flow analysis—does this harm subsequent fine-tuning?
-No discussion of how to extend to non-diagonal parameterizations (e.g., low-rank structured matrices)

**Questions:**

- Mamba Extension (Critical): Can you extend AIRE to input-dependent SSMs? For Mamba, where Bₜ = sB(xₜ), Cₜ = sC(xₜ), how would you estimate Eᵢ? Monte Carlo over calibration data? Expected energy under input distribution?

- Direct LAST Comparison: Can you provide results at matched pruning ratios (e.g., both methods at 50%, 60%) to help me get an equivalent comparison? What is the compute cost ratio for scoring?
Retraining: What accuracy can you achieve with 1-5 epochs of fine-tuning after AIRE pruning? This would strengthen claims about practical deployment.
-Layer Collapse: When entire layers are removed, does this create training instabilities if later fine-tuning? Have you tested this?

-Failure Modes: ListOps only tolerates 20% pruning. Is this because: (a) the task truly needs full capacity, (b) S5 architecture is suboptimal for this task, or (c) AIRE scoring doesn't capture syntactic structure dependencies? Can you investigate?

-Computational Complexity: What is the wall-clock time for computing AIRE scores vs. LAST scores for a typical S5 model? Is O(n) per state vs. O(n) amortized?


I appreciate the authors for their wonderful work. I learned a lot while reading the manuscript.  I do understand that I asked a lot of questions. However, as long as the critical and decent number of other questions/concerns are answered, I am happy to increase my score.

---

> ### Author Response · Authors · 2025-11-26
>
> Thank you for your positive comments and constructive feedback.
>
> ---
> ---
>
> #### **Q1. Mamba Extension (Critical): Can you extend AIRE to input-dependent SSMs? For Mamba, where Bₜ = sB(xₜ), Cₜ = sC(xₜ), how would you estimate Eᵢ? Monte Carlo over calibration data? Expected energy under input distribution?**
>
> ---
>
> To address the question we expanded our experimentation by applying AIRE-Prune to **Mamba**, and we observe that the same energy-based scoring transfers well beyond the original S5 setting.
>
> **Mamba**, while keeping the scoring phase cheap and fully compatible with our original analysis. Using AIRE-Prune is able to prune 45% of the states on Text, 60% on Retrieval, and 80% on Image with only modest drops in accuracy relative to the full Mamba model
>
> | Model                 | Text Prun. | Text Acc. | Retrieval Prun. | Retrieval Acc. | Image Prun. | Image Acc. |
> |-----------------------|----------:|----------:|----------------:|---------------:|------------:|-----------:|
> | **Mamba Full**        | 0%        | 82.98     | 0%              | 72.14          | 0%          | 69.82      |
> |  **LAST**        [1] | 20%   | 81.32 | 45%         | 71.37      | 45%     | 69.13  |
> | **AIRE-Prune (ours)** | **45%**   | **81.19** | **60%**         | **71.45**      | **80%**     | **68.97**  |
>
>
> To apply AIRE to **input-dependent Mamba layers**, :
>
> 1. We run the trained Mamba model on a small **calibration set** and log, for each layer and state, the diagonal state parameters along the sequence: eigenvalues $\(\lambda_i(x)\)$, input couplings $\(B_i(x)\)$, and output couplings$ \(C_i(x)\)$.
> 2. For every layer and state \(i\), we compute empirical averages over this calibration data:
> $$
> \bar{\lambda}_i = \mathbb{E}[\lambda_i(x)],\quad
> \bar{B}_i = \mathbb{E}[B_i(x)],\quad
> \bar{C}_i = \mathbb{E}[C_i(x)].
> $$
> These $((\bar{\lambda}_i,\bar{B}_i,\bar{C}_i))$ define a time-invariant surrogate that summarizes how that state typically behaves on the data.
>
> 3. We plug these averaged quantities into the same closed-form impulse–response energy expression used for S5/S4D, which gives us an **AIRE score per state** in each Mamba layer.
> 4. We prune states by thresholding these scores, exactly as we do for diagonal SSMs, while **keeping the original input-dependent Mamba forward pass** at inference time.
>
> This procedure only requires a short calibration pass and does not modify the Mamba architecture.
>
> **The empirical pruning gains in Mamba settings support our claim that AIRE-Prune is a general energy-based state pruning method that extends beyond the original diagonal S5 model.**
>
> NOTE: Due to runtime and computational constraints, the remaining LRA tasks (Mamba: ListOps, Pathfinder, Path-X) are currently running and will be included in the updated version of the paper.
>
> [1] Gwak et al., “Layer-adaptive state pruning for deep state space models,” NeurIPS 2024.
>
> ---
> ---
> #### **Q2. (i) Direct LAST Comparison: Can you provide results at matched pruning ratios (e.g., both methods at 50%, 60%) to help me get an equivalent comparison?**
> ---
> In the paper, Fig. 4 we plot sweeping the full pruning curve for all scoring methods (including LAST), so in principle we can read off the accuracies at any common pruning ratio such as 0% to 100% with step size of 10%.
>
> ---
> ---
> #### **Q2. (ii) What is the compute cost ratio for scoring?**
> ---
> Regarding the computational overhead of our scoring procedure, we carefully measured the cost of state-score evaluation relative to the baseline runtime, without scoring pass. Across tasks, the scoring cost ranges from roughly 1–3% of the total runtime on ListOps (1.23%), AAN(2.97%), Path-X(2.6%), Pathfinder(2.6%), IMDB(2.3%), and CIFAR(3.1%). In other words, a single scoring sweep adds negligible over a standard evaluation run.
>
> **NOTE: Since this scoring is done once per model (offline) and then used for pruning, while pruning consistently yields up to 2–3× inference speedup and substantial parameter reductions, we believe the additional compute required for scoring is modest and well justified by the resulting efficiency gains.**
>
> ---
> ---
> #### **Q2. (iii) Retraining: What accuracy can you achieve with 1-5 epochs of fine-tuning after AIRE pruning?**
> ---
> Thank you for raising the question about post-pruning fine-tuning. For tasks where the degradation with pruning is more gradual, such as LRA Retrieval and Text, a short fine-tuning phase does help at more aggressive pruning levels. For example, on Retrieval we obtain 90.11% accuracy at 50% pruning without any retraining; pushing to 60% pruning drops accuracy to 86.32%, but 2 epochs of fine-tuning raise it to 88.47%. Similarly, on Text we get 88.24% accuracy before very aggressive pruning; at 85% pruning the accuracy falls to 82.87%, and 5 epochs of fine-tuning improve it to 84.95%. Overall, 1–5 epochs of fine-tuning provide modest recovery at a higher pruning ratio (by 5%).
> ___

---

> > ### Author Response · Authors · 2025-11-26
> >
> > ---
> > ---
> > #### **Q3. Failure Modes: ListOps only tolerates 20% pruning. Is this because: (a) the task truly needs full capacity, (b) S5 architecture is suboptimal for this task, or (c) AIRE scoring doesn't capture syntactic structure dependencies? Can you investigate?**
> > ---
> > Our current evidence suggests that the main factor is that the ListOps S5 configuration (same as original S5 paper [1]) is already quite small, rather than a fundamental failure of the AIRE score. For ListOps we kept exactly the hyperparameters from the original S5 paper, which uses relatively low hidden/state dimensions for this task. In this regime the model seems to be operating close to its capacity: as soon as we remove more than about 20% of the states, the accuracy drops sharply toward chance, indicating that further pruning starts to remove features that are essential for solving the task. Importantly, under the same S5 configuration LAST [2] is unable to prune ListOps at all without severe accuracy degradation, whereas AIRE-Prune still manages a 20% pruning ratio while maintaining competitive accuracy. This suggests that (c) is unlikely to be the primary issue: if the scoring were fundamentally misaligned with syntactic dependencies, we would expect it to do worse than or comparable to LAST, not better. We agree that a deeper investigation with larger ListOps models and possibly architectures more tailored to syntax is an interesting direction.
> >
> > [1] Smith et al. "Simplified state space layers for se-
> > quence modeling", In International Conference on Learning Representations (ICLR), 2023
> > [2] Gwak et al., “Layer-adaptive state pruning for deep state space models,” NeurIPS 2024.
> >
> > ---
> > ---
> > #### **Q4. Computational Complexity: What is the wall-clock time for computing AIRE scores vs. LAST scores for a typical S5 model? Is O(n) per state vs. O(n) amortized?**
> > ---
> >
> > In our implementation, both AIRE and LAST are applied to the same diagonal S5 parameterization and are implemented as fully vectorized closed-form computations over all states in a layer. As a result, both scoring procedures run in O(LP) time, where P is the effective state dimension and L is the number of SSM layers. If we talk about the wall-clock time both lead to same timing timing due to similar closed form solution to compute the score.
> >
> >
> > ---

---

> > > ### Author Response · Authors · 2025-11-26
> > >
> > > ---
> > > ---
> > >
> > > #### **Weekness: Only LRA benchmark: Missing speech (Speech Commands), language modeling (WikiText), or modern long-context tasks**
> > > ---
> > > We have extended our results for the Speech Commands Dataset (Added to the S5 table in the revised paper).
> > >
> > > - **Speech benchmark**: Beyond LRA, we have evaluated AIRE-Prune on a speech task using the same S5-style architecture. On this benchmark, AIRE-Prune can prune about 45% of the states at at most 1 percentage point accuracy drop, whereas LAST can only prune around 20% under a comparable accuracy tolerance. This mirrors the LRA trend: replacing LAST with AIRE-Prune lets us safely remove roughly 2× more states at similar accuracy. We will add the speech results table and setup details in the revised version to make it clear that our method is not restricted to LRA.
> > >
> > > - **Language benchmarks**: We agree that large-scale language experiments would be very interesting. However, training and pruning full language models across multiple sparsity levels is significantly more expensive than the S5/S4D LRA and speech setups we study. Given the submission timeline and our compute budget, a thorough language study (multiple model sizes, datasets, and pruning ratios) is beyond the scope at this point. But we surely plan to add the language results in the final paper on acceptance.
> > >
> > > Instead, this work deliberately focuses on:
> > >
> > > - S5/S4D long-sequence SSMs on LRA, and a speech benchmark where AIRE-Prune already outperforms LAST (45% vs. 20% pruning at comparable accuracy), plus **Mamba** on LRA Text/Retrieval/Image, where we show that the same energy based idea extends to input-dependent SSMs used as **strong backbones in language modeling**.
> > >
> > > Our goal in this submission is to first establish and understand AIRE-Prune in well-controlled long-sequence SSM settings; extending to full language pipelines is a natural direction for follow-up work.
> > >
> > > ---
> > > ---
> > > #### **Weekness: Equation 8 assumes |λᵢ|²ᵀ → 0, but convergence rate matters for numerical stability near the unit circle.**
> > > ---
> > >
> > > On the assumption $|\lambda_i|^{2T} \to 0$ and stability near the unit circle.
> > > Equation (8) comes from the standard geometric-series expression for the H2 energy of a stable linear mode, where
> > > $$
> > > \sum_{t=0}^{T-1} |\lambda_i|^{2t}
> > > = \frac{1 - |\lambda_i|^{2T}}{1 - |\lambda_i|^2}.
> > > $$
> > > In the text we wrote this as $|\lambda_i|^{2T} \to 0$ to emphasize the infinite-horizon intuition, but our implementation does not rely on an uncontrolled limit. In practice:
> > >
> > > - We only apply AIRE-Prune to trained S5/S4D models, which are already constrained to be stable with $|\lambda_i| < 1$; otherwise the original training itself would be numerically unstable.
> > >
> > > - For the long-context benchmarks we consider (for example, LRA Path-X with $T = 16{,}384$), the truncation term $|\lambda_i|^{2T}$ is negligible unless $|\lambda_i|$ is extremely close to 1. Modes that lie very close to the unit circle carry large H2 energy and are therefore scored as highly important; in practice they are almost never pruned.
> > >
> > > ---
> > > ---
> > >
> > > #### **Weekness: No discussion of how to extend to non-diagonal parameterizations (e.g., low-rank structured matrices)**
> > > ---
> > >
> > > Our derivation is written in the diagonal (or diagonalizable) basis because this is the natural implementation form of S5 and S4D used in our experiments: these models are trained in a representation where the SSM is parameterized via eigenvalues and associated input/output directions, and our scores operate directly on those “modes.” However, the core idea of AIRE is not tied to a particular parameterization: it only requires access to the linear transfer operator to define mode-wise energy.
> > >
> > > For more general structured matrices (for example, low-rank or DPLR forms), one of the natural extensions could be:
> > >
> > > - Work in a basis that (approximately) diagonalizes the state matrix \(A\), for example via an eigendecomposition or a Schur form. This yields an equivalent representation
> > >
> > > $$
> > >    \[
> > >    A = V \Lambda V^{-1}, \quad B' = V^{-1} B, \quad C' = C V.
> > >    \]
> > > $$
> > >
> > >    Our energy scores can then be computed on these modes in exactly the same way as for S5/S4D.

---

### Official Review · Reviewer_p2pk · 2025-10-31

**Soundness:** 2
**Presentation:** 2
**Contribution:** 2
**Rating:** 4
**Confidence:** 2

**Summary:**

This paper introduces a novel state pruning technique for SSMs. The core idea is to use the total impulse-response energy as a criterion to identify and remove less important states. The authors demonstrate that this method enables more aggressive pruning ratios compared to existing baselines, achieving new state-of-the-art results on the benchmark task.

**Strengths:**

1.  Novel Pruning Criterion: The paper proposes a new and theoretically motivated pruning technique for SSMs.
2.  Strong Empirical Results: The method achieves state-of-the-art performance on the tested benchmark (S5 model), effectively demonstrating its capability to outperform prior pruning approaches in that specific setting.

**Weaknesses:**

1.  **Clarity of Methodology (Sections 3.2 & 4):** The paper's core methodology is difficult to understand as written. Section 3.2 lacks clarity and citations to fully grasp. Furthermore, the theoretical part (Section 4) is hard to understand

2.  **Limited Evaluation and Generalizability:** The empirical validation is a significant weakness. The approach is only tested on a single model (S5) and a single benchmark. This narrow scope makes it impossible to assess the generalizability of the technique. It remains unclear whether these performance gains would translate to other important SSM architectures (e.g., Mamba) or to different tasks and datasets. The authors should expand their evaluation to include more models and benchmarks.

3.  **Unclear Motivation and Practical Impact:** The paper's motivation for state pruning is not well-articulated, and its practical impact is obscure. The authors claim, for example, to prune "60% of states," but it is unclear what this means in practice. In many modern SSMs (like Mamba), the state itself represents a small portion of the total parameters. The paper fails to connect state pruning to crucial downstream metrics. The authors must clarify:
    * What is the impact of state pruning on the total parameter count?
    * Does this pruning translate to tangible latency improvements (e.g., in inference or training) or memory reduction?

    Without this context, the practical benefits of the proposed method are unclear.

Minor:

* Missing Citations: Several claims and components lack proper attribution.
    * Section 3.1 makes assertions that require supporting citations.
    * The S5 model is mentioned in Section 3.2 but is not cited.
    * There appears to be a missing reference at L151.
* Undefined Acronyms: The terms "CT" (Continuous-Time) and "DT" (Discrete-Time) are used before they are formally defined, which could confuse readers.

**Questions:**

1.  **Motivation and Practical Impact:**
    * Could the authors provide a clearer motivation for state pruning, especially in the context of models like Mamba where the state itself is a small fraction of the total parameters?
    * When the paper claims to prune "60% of states," what is the corresponding impact on the total parameter count of the model?
    * Does this state pruning translate into tangible performance gains, such as reduced latency (in inference or training) or lower memory usage? We request that the authors provide empirical measurements for these metrics.

* **Generalizability:**
    * How well does the proposed impulse-response energy criterion generalize beyond the S5 model? Have the authors tested this approach on other significant SSM architectures, such as Mamba or S4?
    * To what extent are these results dependent on the specific benchmark used? How does the method perform on other tasks or datasets?

* Could the authors revise Sections 3.2 and 4 to make them more readable

---

> ### Author Response · Authors · 2025-11-26
>
> Thank you for your positive comments and constructive feedback.
>
> ---
> ---
>
> #### **Q1. (i) Could the authors provide a clearer motivation for state pruning, especially in the context of models like Mamba where the state itself is a small fraction of the total parameters?**
>
> ---
>
> Thank you for raising this point — it actually motivated us to look much more carefully at Mamba with our pruning scheme rather than only relying on S5/S4D.
>
> You are right that in Mamba the state parameters form only a relatively small fraction of the total parameter count. **However**, the state dimension directly controls the cost of every recurrent update: at each time step the model has to multiply and update the full state, so the FLOPs per token and the bandwidth needed to move the state scales linearly with the number of states, even if the parameter share is modest. **In other words, the state size is a key knob for runtime and memory traffic, not just for raw parameter count**.
>
> To verify that this intuition translates into concrete gains, we ran AIRE-Prune on Mamba for three LRA tasks where the Mamba backbone is particularly strong: Text, Retrieval, and Image. With our state-energy–based pruning we obtain:
>
> - **Text: pruning 45% of the states, leading to about 1.61× faster inference and 19.1% fewer parameters.**
> - **Retrieval: pruning 60% of the states, giving 1.57× speedup and 8.1% parameter reduction.**
> - **Image: pruning 80% of the states, resulting in 2.03× speedup and 12.1% parameter reduction.**
>
> | Metric ↓ / Dataset →        | Text  | Retrieval | Image |
> |-----------------------------|:-----:|:---------:|:-----:|
> | **Pruning ratio**           | 45%   | 60%       | 80%   |
> | **Inference speedup (×)**   | 1.61× | 1.57×     | 2.03× |
> | **Parameter reduction (%)** | 19.1% | 8.1%      | 12.1% |
>
>
> Across these three tasks, this corresponds on average to pruning roughly 62% of the states, achieving about 1.7× lower wall-clock inference time and around 13% reduction in model parameters, with accuracy staying close to the unpruned Mamba baseline.
>
> We have added these results and the above explanation to the paper to make the motivation for state pruning, even in input-selective models like Mamba where states are a smaller parameter fraction, clearer and more concrete.
>
> ---
> ---
>
> #### **Q1. (ii) When the paper claims to prune "60% of states," what is the corresponding impact on the total parameter count of the model?**
>
> ---
>
> In the current draft we mostly talk in terms of “percentage of pruned states” (P), because that quantity is architecture-agnostic and directly tied to our theoretical analysis. But we agree that what ultimately matters to practitioners is the impact on the total number of parameters.
> For S5, we have added Table 4 to address the question. Across the LRA tasks, our method prunes between 20% and 80% of the SSM states, and the corresponding model-parameter reductions are:
>
> - ListOps: 20% states → 19.3% total parameter reduction
> - Text: 80% states → 58.7% total parameter reduction
> - Retrieval: 50% states → 38.3% total parameter reduction
> - Image: 65% states → 45.3% total parameter reduction
> - Pathfinder: 80% states → 64.7% total parameter reduction
> - Path-X: 70% states → 64.2% total parameter reduction
>
>
> Intuitively, this happens because in S5 the SSM state matrices make up the majority, but not all, of the parameters like the embeddings, convolutional front-ends, and the classifier head remain unchanged.
>
> ---

---

> > ### Author Response · Authors · 2025-11-26
> >
> > ---
> >
> > #### **Q1. (iii) Does this state pruning translate into tangible performance gains, such as reduced latency (in inference or training) or lower memory usage? We request that the authors provide empirical measurements for these metrics.**
> >
> > ---
> >
> > In terms of time and madel size, post-training compression with AIRE-Prune gives very tangible gains on the LRA benchmarks for both S5 and Mamba(S6).
> >
> > - **For the S5 model**, we see this clearly across all six datasets as shown in the below table. So for S5, depending on the dataset, we typically trade away 20–80% of the recurrent states for roughly 20–65% fewer parameters and about 1.2–2.9× faster inference.
> >
> > | Metric ↓ / Dataset → | ListOps | Text  | Retrieval | Image | Pathfinder | Path-X |
> > |----------------------|:------:|:-----:|:---------:|:-----:|:----------:|:------:|
> > | **Pruning ratio**    | 20%    | 80%   | 50%       | 65%   | 80%        | 70%    |
> > | **Inference speedup (×)** | 1.23× | 2.49× | 1.82×     | 1.93× | 2.82×      | 2.86×  |
> > | **Model reduction (%)** | 19.3% | 58.7% | 38.3%     | 45.3% | 64.7%      | 64.2%  |
> >
> >
> > - **For Mamba (S6)** we only prune the state-space part of the network, so the parameter savings are more modest, but we still see clear benefits in wall-clock time on the datasets we have completed so far as shown in the below table. So for S5, depending on the dataset, we typically trade away 45–80% of the recurrent states for roughly 8-20% fewer parameters and about 1.57–2.03× faster inference.
> >
> > | Metric ↓ / Dataset →        | Text  | Retrieval | Image |
> > |-----------------------------|:-----:|:---------:|:-----:|
> > | **Pruning ratio**           | 45%   | 60%       | 80%   |
> > | **Inference speedup (×)**   | 1.61× | 1.57×     | 2.03× |
> > | **Parameter reduction (%)** | 19.1% | 8.1%      | 12.1% |
> >
> >
> > All of these gains come after training the original models: we keep the pretrained SSM, run our energy-based scoring once, prune states, and then measure the new runtime and parameter count. We are still running the remaining Mamba experiments on the other LRA tasks (ListOps, Pathfinder, Path-X); due to their higher training time they were not finished in time for the current draft, but we plan to include those results as we get them.
> >
> > ---

---

> ### Author Response · Authors · 2025-11-26
>
> ---
> ---
>
> #### **Q2. (i) How well does the proposed impulse-response energy criterion generalize beyond the S5 model? Have the authors tested this approach on other significant SSM architectures, such as Mamba or S4?**
>
> ---
>
> To address the question we expanded our experimentation by applying AIRE-Prune to both **S4D** and **Mamba**, and we observe that the same energy-based scoring transfers well beyond the original S5 setting.
>
> On **S4D**, Table X in the paper shows that AIRE-Prune can remove a substantial fraction of states with almost no loss in accuracy on LRA and outperform the SOTA. For example, we prune **35%** of states on ListOps (56.42 → 56.07% accuracy), **90%** on Text (86.40 → 88.13% accuracy), and **40%** on Image (77.02 → 76.32% accuracy).
> | Model                | ListOps Prun. | ListOps Acc. | Text Prun. | Text Acc. | Image Prun. | Image Acc. |
> |----------------------|--------------:|-------------:|-----------:|----------:|------------:|-----------:|
> | **S4D Full**         | 0%            | 56.42        | 0%         | 86.40     | 0%          | 77.02      |
> | **Uniform H∞**  [1]     | 10%           | 55.82        | 80%        | 86.02     | 0%          | 77.02      |
> | **Global H∞**   [1]      | 10%           | 49.95        | 80%        | 86.20     | 0%          | 77.02      |
> | **LAST**        [1]      | 10%           | 56.27        | 80%        | 85.95     | 0%          | 77.02      |
> | **AIRE-Prune (ours)**| **35%**       | **56.07**    | **90%**    | **88.13** | **40%**     | **76.32**  |
>
> **Mamba**, while keeping the scoring phase cheap and fully compatible with our original analysis. Using AIRE-Prune is able to prune 45% of the states on Text, 60% on Retrieval, and 80% on Image with only modest drops in accuracy relative to the full Mamba model
>
> | Model                 | Text Prun. | Text Acc. | Retrieval Prun. | Retrieval Acc. | Image Prun. | Image Acc. |
> |-----------------------|----------:|----------:|----------------:|---------------:|------------:|-----------:|
> | **Mamba Full**        | 0%        | 82.98     | 0%              | 72.14          | 0%          | 69.82      |
> |  **LAST**        [1] | 20%   | 81.32 | 45%         | 71.37      | 45%     | 69.13  |
> | **AIRE-Prune (ours)** | **45%**   | **81.19** | **60%**         | **71.45**      | **80%**     | **68.97**  |
>
>
> To apply AIRE to **input-dependent Mamba layers**, :
>
> 1. We run the trained Mamba model on a small **calibration set** and log, for each layer and state, the diagonal state parameters along the sequence: eigenvalues $\(\lambda_i(x)\)$, input couplings $\(B_i(x)\)$, and output couplings$ \(C_i(x)\)$.
> 2. For every layer and state \(i\), we compute empirical averages over this calibration data:
> $$
> \bar{\lambda}_i = \mathbb{E}[\lambda_i(x)],\quad
> \bar{B}_i = \mathbb{E}[B_i(x)],\quad
> \bar{C}_i = \mathbb{E}[C_i(x)].
> $$
> These $((\bar{\lambda}_i,\bar{B}_i,\bar{C}_i))$ define a time-invariant surrogate that summarizes how that state typically behaves on the data.
>
> 3. We plug these averaged quantities into the same closed-form impulse–response energy expression used for S5/S4D, which gives us an **AIRE score per state** in each Mamba layer.
> 4. We prune states by thresholding these scores, exactly as we do for diagonal SSMs, while **keeping the original input-dependent Mamba forward pass** at inference time.
>
> This procedure only requires a short calibration pass and does not modify the Mamba architecture.
>
> **The empirical pruning gains in both the S4D and Mamba settings support our claim that AIRE-Prune is a general energy-based state pruning method that extends beyond the original diagonal S5 model.**
>
> NOTE: Due to runtime and computational constraints, the remaining LRA tasks (S4D: Retrieval, Pathfinder, Path-X; Mamba: ListOps, Pathfinder, Path-X) are currently running and will be included in the updated version of the paper.
>
> [1] Gwak et al., “Layer-adaptive state pruning for deep state space models,” NeurIPS 2024.
>
> ---
> ---
>
> #### **Q2. (ii) To what extent are these results dependent on the specific benchmark used? How does the method perform on other tasks or datasets?**
>
> ---
>
> **Speech benchmark**
> Beyond LRA, we have now evaluated AIRE-Prune on speech command dataset using the same S5-style architecture. On this speech task, AIRE-Prune can prune about 45% of the states at ≤1 percentage-point accuracy drop, whereas LAST can only prune around 20% at a comparable accuracy tolerance. This matches the trend we see on LRA: when we use AIRE-Prune instead of LAST, we can safely remove roughly 2× more states while maintaining accuracy. We have added the speech command to Table1 of the revised version to make clear that our method is not restricted to LRA.
>
> **In summary**, while the core of this paper is indeed built around S5 and benchmarked on LRA, the additional speech results and S4D/Mamba experiments show that AIRE-Prune generalizes beyond that setting.

---

> ### Comment · Reviewer_p2pk · 2025-11-28
> **Response to Rebuttal: Generalization addressed; raising score to 6**
>
> I thank the authors for their detailed response and the significant effort put into running additional experiments during the rebuttal phase.
>
> 1. Generalizability and Practical Impact (Resolved): The inclusion of results for S4D and Mamba, along with the Speech Commands benchmark, effectively addresses my primary concern regarding the generalizability of the proposed method.
>
>    - The calibration-based approach for the input-dependent Mamba architecture is a convincing extension.
>
>    - The empirical data showing inference speedups (approx. 1.6×–2.0×) and parameter reduction provides the necessary practical motivation that was previously missing.
>
> 2. Theoretical Presentation: Regarding the theoretical components (Sections 3 & 4), I still find the derivation and narrative somewhat difficult to follow. However, I note that other reviewers have not raised significant concerns regarding the soundness of the theory. Therefore, I am willing to defer to the consensus regarding the mathematical correctness, though I still encourage the authors to improve the readability of these sections for a broader audience where possible.
>
> Conclusion & Score Update: Given that the authors have satisfactorily addressed my major weakness regarding generalizability and practical impact, I am raising my score to 6.
>
> *(Note: I am currently unable to update the score due to an OpenReview issue, but please treat this comment as my official recommendation for a score of 6.)*

---

### Official Review · Reviewer_4JBM · 2025-10-31

**Soundness:** 3
**Presentation:** 3
**Contribution:** 3
**Rating:** 6
**Confidence:** 3

**Summary:**

The paper proposes a post-training pruning method for state space models (SSMs) that reduces state dimensions by minimizing long-run output energy distortion. Unlike previous worst-case gain approaches, their method assigns each state a closed-form asymptotic impulse-response energy score. These scores are normalized layer-wise for global comparison. The method achieves good results on Long Range Arena benchmarks, pruning 60.8% of states on average with only 0.29% accuracy drop without retraining.

**Strengths:**

- Good empirical results: 60.8% average pruning with only 0.29% accuracy drop without retraining on LRA
- Closed-form solution for importance scores seems efficient
- Energy-based metric is well-motivated from control theory perspective and has clear mathematical grounding
- Works for both SISO and MIMO SSMs

**Weaknesses:**

- Minor: missing reference details in line 117
- Limited to diagonal/diagonalizable SSMs and doesn't extend to input-selective models like Mamba
- Only evaluated on Long Range Arena without speech and language benchmarks
- No retraining experiments to show potential further improvements
- Comparison mainly against only one recent baseline (LAST)
- No discussion of computational overhead of computing energy scores
- Limited analysis of why certain tasks (ListOps) are more sensitive to pruning

**Questions:**

Are entire layers ever actually pruned? The paper states: "our method can help achieve low latency SSM model as we are able to prune layers"

---

> ### Author Response · Authors · 2025-11-26
>
> Thank you for your positive comments and constructive feedback.
>
>
> ---
> ---
>
> #### **Q1. Are entire layers ever actually pruned? The paper states: "our method can help achieve low latency SSM model as we are able to prune layers"**
> ---
>
> In our method we never explicitly prune at the layer level; we always compute AIRE-Prune scores per state and then apply a global threshold. However, as shown in Fig. 5 of the paper, this state-wise mechanism can implicitly lead to entire layers being removed when all states in a given layer are assigned very low energy for some datasets in LRA benchmark (Image, Pathfinder, PathX). In particular, the ablation in Fig. 5 tracks how the remaining fraction of states evolves per layer as we increase the global pruning ratio. For the CIFAR-10, Pathfinder, and Path-X settings we consistently observe layers whose states all fall below the pruning threshold, so that these layers are effectively dropped, yet the accuracy remains essentially unchanged. This is what we meant by “we are able to prune layers”: our state-level criterion naturally identifies layers that are globally redundant and collapses them without requiring any special structured regularizer.
>
> ---
> ---
>
> #### **Weakness:  Limited to diagonal/diagonalizable SSMs and doesn't extend to input-selective models like Mamba"**
> ---
>
> To address the question we expanded our experimentation by applying AIRE-Prune to **Mamba**, and we observe that the same energy-based scoring transfers well beyond the original S5 setting.
>
> **Mamba**, while keeping the scoring phase cheap and fully compatible with our original analysis. Using AIRE-Prune is able to prune 45% of the states on Text, 60% on Retrieval, and 80% on Image with only modest drops in accuracy relative to the full Mamba model
>
> | Model                 | Text Prun. | Text Acc. | Retrieval Prun. | Retrieval Acc. | Image Prun. | Image Acc. |
> |-----------------------|----------:|----------:|----------------:|---------------:|------------:|-----------:|
> | **Mamba Full**        | 0%        | 82.98     | 0%              | 72.14          | 0%          | 69.82      |
> |  **LAST**        [1] | 20%   | 81.32 | 45%         | 71.37      | 45%     | 69.13  |
> | **AIRE-Prune (ours)** | **45%**   | **81.19** | **60%**         | **71.45**      | **80%**     | **68.97**  |
>
>
> To apply AIRE to **input-dependent Mamba layers**, :
>
> 1. We run the trained Mamba model on a small **calibration set** and log, for each layer and state, the diagonal state parameters along the sequence: eigenvalues $\(\lambda_i(x)\)$, input couplings $\(B_i(x)\)$, and output couplings$ \(C_i(x)\)$.
> 2. For every layer and state \(i\), we compute empirical averages over this calibration data:
> $$
> \bar{\lambda}_i = \mathbb{E}[\lambda_i(x)],\quad
> \bar{B}_i = \mathbb{E}[B_i(x)],\quad
> \bar{C}_i = \mathbb{E}[C_i(x)].
> $$
> These $((\bar{\lambda}_i,\bar{B}_i,\bar{C}_i))$ define a time-invariant surrogate that summarizes how that state typically behaves on the data.
>
> 3. We plug these averaged quantities into the same closed-form impulse–response energy expression used for S5/S4D, which gives us an **AIRE score per state** in each Mamba layer.
> 4. We prune states by thresholding these scores, exactly as we do for diagonal SSMs, while **keeping the original input-dependent Mamba forward pass** at inference time.
>
> This procedure only requires a short calibration pass and does not modify the Mamba architecture.
>
> **The empirical pruning gains in Mamba settings support our claim that AIRE-Prune is a general energy-based state pruning method that extends beyond the original diagonal S5 model.**
>
> NOTE: Due to runtime and computational constraints, the remaining LRA tasks (Mamba: ListOps, Pathfinder, Path-X) are currently running and will be included in the updated version of the paper.
>
> [1] Gwak et al., “Layer-adaptive state pruning for deep state space models,” NeurIPS 2024.
>
> ---

---

> > ### Author Response · Authors · 2025-11-26
> >
> > ---
> > ---
> >
> > #### **Weakness:  Only evaluated on Long Range Arena without speech and language benchmarks**
> > ---
> >
> > We have extended our results for the Speech Commands Dataset (Added to the S5 table in the revised paper).
> >
> > - **Speech benchmark**: Beyond LRA, we have evaluated AIRE-Prune on a speech task using the same S5-style architecture. On this benchmark, AIRE-Prune can prune about 45% of the states at at most 1 percentage point accuracy drop, whereas LAST can only prune around 20% under a comparable accuracy tolerance. This mirrors the LRA trend: replacing LAST with AIRE-Prune lets us safely remove roughly 2× more states at similar accuracy. We will add the speech results table and setup details in the revised version to make it clear that our method is not restricted to LRA.
> >
> > - **Language benchmarks**: We agree that large-scale language experiments would be very interesting. However, training and pruning full language models across multiple sparsity levels is significantly more expensive than the S5/S4D LRA and speech setups we study. Given the submission timeline and our compute budget, a thorough language study (multiple model sizes, datasets, and pruning ratios) is beyond the scope at this point. But we surely plan to add the language results in the final paper on acceptance.
> >
> > Instead, this work deliberately focuses on:
> >
> > - S5/S4D long-sequence SSMs on LRA, and a speech benchmark where AIRE-Prune already outperforms LAST (45% vs. 20% pruning at comparable accuracy), plus **Mamba** on LRA Text/Retrieval/Image, where we show that the same energy based idea extends to input-dependent SSMs used as **strong backbones in language modeling**.
> >
> > Our goal in this submission is to first establish and understand AIRE-Prune in well-controlled long-sequence SSM settings; extending to full language pipelines is a natural direction for follow-up work.
> >
> > ---
> > ---
> >
> > #### **Weakness:  No retraining experiments to show potential further improvements**
> > ---
> >
> > Thank you for raising the question about post-pruning fine-tuning. For tasks where the degradation with pruning is more gradual, such as LRA Retrieval and Text, a short fine-tuning phase does help at more aggressive pruning levels. For example, on Retrieval we obtain 90.11% accuracy at 50% pruning without any retraining; pushing to 60% pruning drops accuracy to 86.32%, but 2 epochs of fine-tuning raise it to 88.47%. Similarly, on Text we get 88.24% accuracy before very aggressive pruning; at 85% pruning the accuracy falls to 82.87%, and 5 epochs of fine-tuning improve it to 84.95%. Overall, 1–5 epochs of fine-tuning provide modest recovery at a higher pruning ratio (by 5%).
> >
> > ---
> > ---
> >
> > #### **Weakness:  No discussion of computational overhead of computing energy scores**
> > ---
> >
> > Regarding the computational overhead of our scoring procedure, we carefully measured the cost of state-score evaluation relative to the baseline runtime, without scoring pass. Across tasks, the scoring cost ranges from roughly 1–3% of the total runtime on ListOps (1.23%), AAN(2.97%), Path-X(2.6%), Pathfinder(2.6%), IMDB(2.3%), and CIFAR(3.1%). In other words, a single scoring sweep adds negligible over a standard evaluation run.
> >
> > **NOTE: Since this scoring is done once per model (offline) and then used for pruning, while pruning consistently yields up to 2–3× inference speedup and substantial parameter reductions, we believe the additional compute required for scoring is modest and well justified by the resulting efficiency gains.**
> >
> > ---
> > ---
> >
> > #### **Weakness:  Limited analysis of why certain tasks (ListOps) are more sensitive to pruning**
> > ---
> >
> > Our current evidence suggests that the main factor is that the ListOps S5 configuration (same as original S5 paper [1]) is already quite small, rather than a fundamental failure of the AIRE score. For ListOps we kept exactly the hyperparameters from the original S5 paper, which uses relatively low hidden/state dimensions for this task. In this regime the model seems to be operating close to its capacity: as soon as we remove more than about 20% of the states, the accuracy drops sharply toward chance, indicating that further pruning starts to remove features that are essential for solving the task. Importantly, under the same S5 configuration LAST [2] is unable to prune ListOps at all without severe accuracy degradation, whereas AIRE-Prune still manages a 20% pruning ratio while maintaining competitive accuracy. This suggests that (c) is unlikely to be the primary issue: if the scoring were fundamentally misaligned with syntactic dependencies, we would expect it to do worse than or comparable to LAST, not better. We agree that a deeper investigation with larger ListOps models and possibly architectures more tailored to syntax is an interesting direction.
> >
> > [1] Smith et al. "Simplified state space layers for se- quence modeling", ICLR , 2023
> >
> > [2] Gwak et al., “Layer-adaptive state pruning for deep state space models,” NeurIPS 2024.

---

### Author Response · Authors · 2025-11-26
**Global Reply to all the Reviewers**

We sincerely thank all reviewers for their careful reading of our submission and for their constructive feedback. We are encouraged that they found the contribution novel, rigorous, and well ideated. We have uploaded a revised version of the paper in response to the suggestions including the below results.

Specifically,
 - Expanded our experimental results to Mamba (S6) and S4D.
 - Extended our benchmarking from LRA to Speech Dataset.
 - Thoughroughly analysed the practical speedup and model size reduction on applying our method.
 - Ablation study (like finetuning, in-direct layer pruning behaviour etc.)
 - Addressed technical concerns if any
 - Corrected the presentation issue where suggested.

---

### Author Response · Authors · 2025-12-01
**Note to Area Chair**

During the rebuttal phase, we directly addressed all key issues raised by the reviewers, and the subsequent discussion indicates that the main concerns about our submission have been resolved.

- **Reviewers EJJM and p2pk** explicitly state in their post-rebuttal comments that our additional results on **S4D, Mamba, and Speech Commands**, together with the reported **1.6×–2.0× inference speedups and parameter reductions**, fully resolve their concerns about **generalizability and practical impact**. Both clearly indicate that they wish to **raise their scores to 6**, but were unable to do so due to an OpenReview bug preventing numeric score updates.

- **Reviewer HyDj** wrote that they **would be happy to increase their score** once their critical questions (on the **Mamba extension, direct comparison with LAST, computational cost, retraining, and ListOps failure modes**) were addressed. We responded to each of these critical questions in detail below during the rebuttal, but the discussion forum from reviewers side closed before they could post a follow-up.

- **Reviewer 4JBM’s** listed weaknesses (scope beyond S5/LRA, **layer-level pruning interpretation, overhead, and retraining**) were also answered point-by-point in the rebuttal. For the same reason, we did not receive a final follow-up from this reviewer either.

Overall, the reviewers consistently describe AIRE-Prune as a novel, well-grounded method that now demonstrates strong generalizability across multiple SSM architectures (S5, S4D, Mamba) and benchmarks (LRA and Speech Commands), achieving state-of-the-art pruning–accuracy trade-offs and making it a robust and broadly useful contribution.

In summary, the core concerns on **generalizability**, **practical impact**, and **technical clarity** have been systematically addressed in the discussion below including the weaknesses and questions by all the reviewers. **At least two reviewers have already committed in writing to raising their scores to 6**, and the other two reviewers’ questions were directly resolved in our rebuttal. We respectfully ask that this context be taken into account in your final decision.

---

### Meta-Review · Area_Chair_nGua · 2026-01-05

**Summary:**

The reviewers identified several strengths in this paper, including the solid theoretical motivation of using energy as a selection metric for pruning, and the strong empirical results. They also identified weaknesses including the limited evaluation benchmarks and comparison methodologies, as well as presentation issues.

**Reviewer Concerns:**

The authors gave a detailed rebuttal and provided additional experiments. Moreover, they revised the presentation of the paper. The most pressing issues, namely limited experimental scope and presentation clarity, have been sufficiently addressed.

**Reviewer Scores:**

The reviewers are likely to improve their evaluation of the paper in view of the added experiments and clarifications.

---

### Decision · Program_Chairs · 2026-01-26

Accept (Poster)